# *From Objects to Anywhere*: A Holistic Benchmark for Multi-level Visual Grounding in 3D Scenes

**Tianxu Wang**[1*]   **Zhuofan Zhang**[1,2*]   **Ziyu Zhu**[1,2]   **Yue Fan**[1]
**Jing Xiong**[1,3]   **Pengxiang Li**[1,4]   **Xiaojian Ma**[1]   **Qing Li**[1†]

[1]State Key Laboratory of General Artificial Intelligence, BIGAI
[2]Tsinghua University [3]Peking University [4]Beijing Institute of Technology

{wangtianxu,liqing}@bigai.ai

**Project page:** https://anywhere-3d.github.io

| Area Level | Space Level | Object Level | Part Level |
|---|---|---|---|

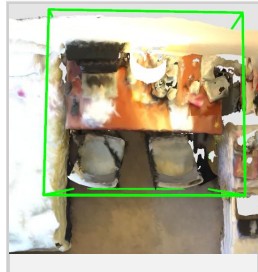 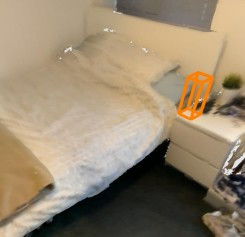 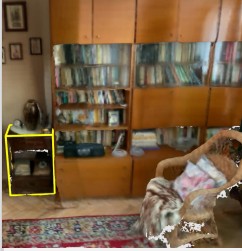 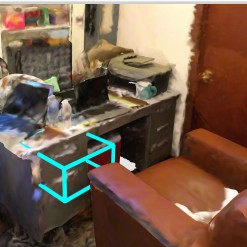

Choose an area that facilitates discussion and allows two people to complete the work efficiently.

Lie on the bed, and place a cup with a diameter of 0.1 meters and a height of 0.3 meters at the upper right corner of the bed table.

Standing up from the armchair and walking 1 meter forward, which object on the floor directly to my right is against the wall?

Assume you are working at your desk. Please pull the top drawer on the left side out as far as it will go until it touches the armchair.

Figure 1: Multi-level Visual grounding in 3D Scenes: area, space, object, and part. Examples illustrate visual grounding of daily life expressions, from functional **areas** for collaborative study, to placing a cup on a nightstand in unoccupied **space**, referring to an **object** via its spatial distance from the armchair, or moving **part** of an object such as pulling out a drawer from a cabinet.

## Abstract

3D visual grounding has made notable progress in localizing objects within complex 3D scenes. However, grounding referring expressions beyond objects in 3D scenes remains unexplored. In this paper, we introduce **Anywhere3D-Bench**, a holistic 3D visual grounding benchmark consisting of 2,886 referring expression-3D bounding box pairs spanning four different grounding levels: human-activity *areas*, unoccupied *space* beyond objects, individual *objects* in the scene, and fine-grained object *parts*. We assess a range of state-of-the-art 3D visual grounding methods alongside large language models (LLMs) and multimodal LLMs (MLLMs) on Anywhere3D-Bench. Experimental results reveal that space-level and part-level visual grounding pose the greatest challenges: space-level tasks require a more comprehensive spatial reasoning ability, for example, modeling distances and spatial relations within 3D space, while part-level tasks demand fine-grained perception of object composition. Even the best performance model, OpenAI o4-mini, achieves only 23.00% accuracy on space-level tasks and 31.46% on part-level tasks, significantly lower than its performance on area-level and object-level tasks. These findings underscore a critical gap in current models' capacity to understand and reason about 3D scenes beyond object-level semantics.

---

[*]Equal contribution
[†]Corresponding author

39th Conference on Neural Information Processing Systems (NeurIPS 2025) Track on Datasets and Benchmarks.

# 1 Introduction

When instructed to place a floor lamp next to an armchair, humans can visually ground it in the scene, estimating its base diameter and height, imagining its precise alignment with the armchair, and judging whether it fits naturally within the 3D environment. Humans can naturally perceive, reason about, and localize expressions to "anywhere" in 3D scenes. Yet can today's 3D vision–language models ground free-form referring expressions to precise positions and dimensions in a 3D scene, especially when those expressions refer to regions beyond objects?

Existing 3D visual grounding models, pretrained on large 3D scene datasets, excel at aligning expressions to objects in a scene [7, 58, 2, 63, 61, 26]. However, these models remain constrained to object-level alignment, with limited attention paid to the broader spatial regions beyond objects. Meanwhile, with the rapid development of Multimodal Large Language Models (MLLMs), an increasing number of studies have begun to explore their ability to perceive and reason about spatial intelligence from 2D images or videos [13, 6, 51, 45, 54, 17, 18, 66]. However, their ability to predict the positions and sizes of 3D bounding boxes corresponding to free-form referring expressions anywhere in 3D space, including both objects and regions beyond object boundaries, remains largely unexplored.

To bridge this gap, we introduce *Anywhere3D-Bench*, a holistic benchmark with 2,886 referring expression-3D bounding box pairs, categorized into four visual grounding levels: area, space, object, and part, as illustrated in Fig. 1. To the best of our knowledge, we are the first to propose a 3D visual grounding benchmark that spans four hierarchical levels of grounding granularity, particularly on aligning expressions with 3D locations and sizes at **space level**. More diverse and illustrative examples can be found in Fig. 2. At each level, we design distinct types of referring expressions to evaluate models' abilities to perceive and reason about various aspects of 3D scene.

We conduct experiments on three categories of models on *Anywhere3D-Bench*: (1) LLMs with textual inputs , (2) MLLMs with both visual and textual inputs, and (3) 3D visual grounding specialist models. Evaluation results reveal that current models perform poorly on our benchmark, particularly on space-level and part-level tasks. Space-level tasks require modeling spatial relationships and distances in unoccupied space beyond individual objects, while part-level tasks demand first identifying the relevant object and then reasoning over its fine-grained structure to predict the appropriate bounding box size and position. Among all models, o4-mini—a strong MLLM with visual reasoning capabilities—achieves the best performance, yet still records only 23.00% accuracy on space-level and 31.46% on part-level tasks. These results are significantly lower than its performance on the object-level (55.82%) and area-level (76.19%) tasks.

With visual inputs from video frames of the scene as well as bird's-eye view image, MLLMs outperform their non-visual LLM counterparts, particularly on object-level and part-level tasks, where detailed visual cues, such as object appearance and structure, can be leveraged. In contrast, gains at the space level are limited, suggesting that spatial relational reasoning in 3D space remains a significant bottleneck for MLLMs. Notably, LLMs and MLLMs generally outperform 3D visual grounding specialist models, especially on space-level tasks. This advantage can be attributed to their pretraining on large-scale image-text datasets, which endows them with stronger perceptual and understanding capabilities. Furthermore, their exposure to real-world knowledge enables a degree of commonsense reasoning, allowing them to infer 3D locations even beyond objects in the scene.

At the object level, our benchmark specifically assesses models' ability to understand quantitative object sizes and inter-object distances, which are rarely emphasized in previous 3D visual grounding benchmarks. The best-performing 3D visual grounding model, Chat-Scene [24], achieves only 31.73% accuracy on the object-level task, substantially lower than the over 50% accuracy reported on benchmarks such as ScanRefer [7], highlighting current 3D visual grounding models' limitations in reasoning about precise object dimensions and spatial distance.

To improve models' performance on the two most challenging grounding levels: space-level and part-level, we introduce the following input enhancements as initial attempts: (1) incorporate global coordinate information and object orientation to support a better understanding of spatial relationships, (2) select key video frames that convey critical visual cues aligned with the referring expression. While these enhancements lead to performance improvements on both space-level and part-level tasks, the gap with human performance remains substantial, highlighting that multi-level visual grounding demands more comprehensive perceptual and reasoning capabilities than current models possess.

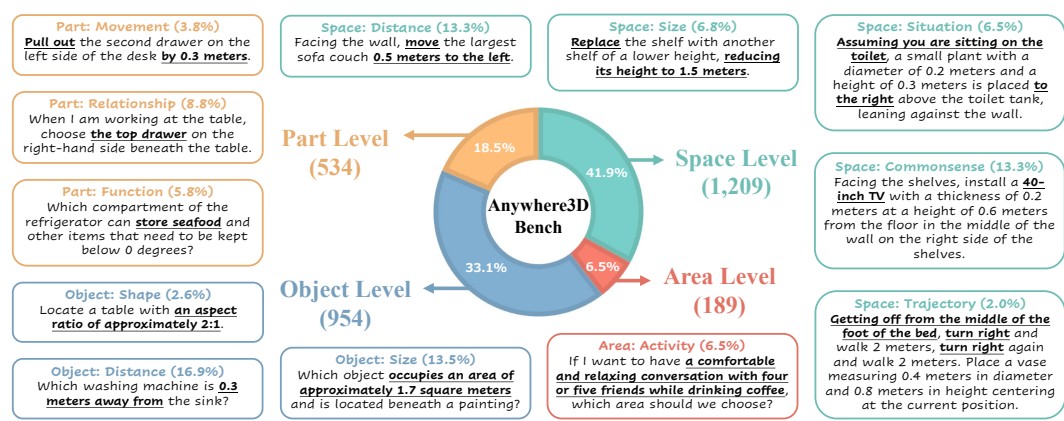

Figure 2: Multi-level visual grounding (Part, Space, Object, Area) with distinct expression types. **Emphasized segments** highlight phrases aligned with their respective expression type.

To summarize, our contributions are as follows:

- We introduce *Anywhere3D-Bench*, the first benchmark for multi-level 3D visual grounding that extends beyond the object level to cover four grounding levels in 3D scenes: area, space, object, and part.

- Experiments on *Anywhere3D-Bench* reveal that space-level and part-level visual grounding are the most challenging tasks. Even the best-performing model, o4-mini, with visual reasoning ability, struggles with these two tasks. Furthermore, compared to MLLMs, 3D visual grounding specialist models exhibit limited performance and poor generalization to space-level tasks.

- Additional spatial and visual cues boost performance on the two most challenging tasks: space-level and part-level. However, a significant gap still remains compared to human performance.

## 2 Anywhere3D Benchmark

Table 1: Comparison of Anywhere3D with existing visual grounding benchmarks (test splits). Anywhere3D expands grounding level to **area**, **space**, **object**, and **part**.

| Benchmark | Area | Space | Object | Part | Test Source | Quality Check | # Scene | # Expression |
|---|---|---|---|---|---|---|---|---|
| ScanRefer [7] | ✗ | ✗ | ✓ | ✗ | Human | ✓ | 97 | 5,410 |
| Nr3D [1] | ✗ | ✗ | ✓ | ✗ | Human | ✓ | 130 | 7,805 |
| Sr3D [1] | ✗ | ✗ | ✓ | ✗ | Template | ✓ | 255 | 17,726 |
| MMScan [35] | ✓ | ✗ | ✓ | ✗ | GPT-4 | ✓ | 702 | 19,696 |
| SceneFun3D [15] | ✗ | ✗ | ✗ | ✓ | Human & Rephrasing | ✓ | 85 | 1,265 |
| ScanReason [63] | ✗ | ✗ | ✓ | ✗ | GPT-4 | ✓ | - | 1,474 |
| **Anywhere3D (ours)** | ✓ | ✓ | ✓ | ✓ | GPT-4 | ✓ | 276 | 2,886 |

As presented in Table 1, we introduce *Anywhere3D-Bench*, which consists of 2,886 referring expression-3D bounding box pairs derived from 276 scenes from the validation sets of ScanNet [14], MultiScan [37], 3RScan [44], and ARKitScenes [4]. Inspired by how people refer to 3D scenes in everyday scenarios, we design four levels of grounding granularity and generate referring expressions specifically tailored to each level: Area Level (189), Space Level (1,209), Object Level (954), and Part Level (534). At each level, we design distinct types of referring expressions aiming at evaluating the models' comprehensive capabilities, as further elaborated in the following section.

### 2.1 Multi-level Visual Grounding

We present the data distribution of *Anywhere3D-Bench* in Fig. 2, along with the representative examples of different types of referring expressions for each level. For detailed benchmark data analysis, please refer to Appendix Section A.1.

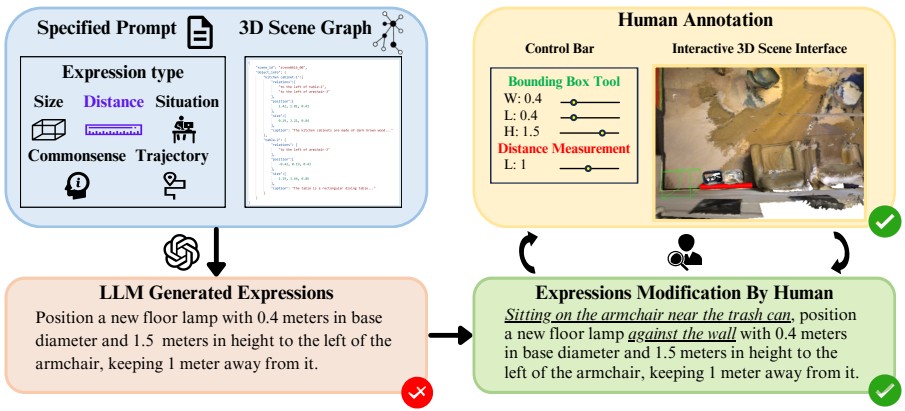

Figure 3: Data generation pipeline of Anywhere3D-Bench. We design specific prompts for different types of expression in four grounding levels. Human annotators are required to annotate the 3D bounding box and refine the GPT-generated expression until they precisely match.

**Area Level**   Expressions belonging to the area level typically describe an indoor ***Activity***, which requires the model to infer the related functional area composed of multiple objects and the space between them.

**Space Level**   Expressions in space level refer to spatial regions beyond objects in the 3D scene and are categorized into the following five types: ***Size***: Expressions that require directly adjusting the size of an object or performing transformations on its dimension (e.g., length, width, height). This category evaluates models' ability to interpret and manipulate quantitative object dimensions in 3D space. ***Distance***: Expressions involving the relocation of an object or the placement of a new object at a specified distance from another. These tasks test models' ability to reason about quantitative spatial distances and relationships. ***Situation***: Expressions involve imagining a scenario in a 3D scene from an egocentric perspective, as introduced in SQA3D [36]. This category evaluates models' ability to understand the situation context and perspective within 3D environments. ***Commonsense***: Expressions that include commonsense knowledge about object size (e.g., "40-inch TV") or typical spatial locations in a scene (e.g., "room corner"). ***Trajectory***: Expressions that specify the starting point and path of a trajectory, requiring the model to place an object at the endpoint of the trajectory and predict the location and size of the object.

**Object Level**   Expressions refer to objects in the 3D scene, following the same setting as in prior works, such as ScanRefer and Nr3D [7, 1]. However, we place particular emphasis on models' ability in reasoning about the quantitative understanding of object ***Size***, ***Shape***, and inter-object ***Distance***.

**Part Level**   Expressions refer to specific parts of objects in a 3D scene and can be categorized into the following three types: ***Movement*** requires models to predict the bounding box of an object part after certain movement, while ***Relationship*** and ***Function*** require models to predict a specific part of an object based on its spatial relationship or function.

## 2.2   Data Generation Pipeline

As shown in Fig. 3, the data generation pipeline of *Anywhere3D-Bench* involves referring expression generation using LLM guided by human-written prompts, along with iterative human annotation and verification. Notably, our annotation interface supports resizing and moving 3D bounding boxes, as well as a distance measurement tool, together enabling precise annotation anywhere in the 3D space.

**Referring Expressions Generation**   To enhance the diversity of referring expressions across various 3D scenes, we leverage GPT-4o [27] to generate expressions regarding different grounding levels as well as different types of expressions in each level. For each scene, we first generate a 3D scene graph following SceneVerse [29]. Each scene graph contains ground-truth object labels, IDs, and 3D bounding boxes of the objects, as well as object captions and inter-object relationships. We then prompt GPT-4o to generate referring expressions by providing the scene graph along with human-written prompts corresponding to a particular expression type and grounding level.

**Human Annotation and Verification**    To ensure the quality of the benchmark, we construct a human-in-the-loop annotation and verification workflow. Annotators are provided with visualizations of the 3D scene as well as ground-truth object labels and sizes, which is adapted from ScanRefer's annotation design. They are allowed to revise the referring expressions and are required to annotate the corresponding 3D bounding boxes via an interactive interface equipped with a bounding box editor and distance measurement tool. A key requirement is emphasized throughout the workflow: **Each referring expression must be grounded exactly to one target 3D bounding box in the scene without ambiguity.**

All annotated expressions and 3D bounding boxes are subsequently verified by humans. Any annotation that does not meet the quality criteria is rejected and iteratively revised until it fully complies with the requirements. Please refer to Appendix Section A.2 and Section A.3 for additional information.

## 3    Experiments and Results

### 3.1    Experimental Setting

**Evaluation Metric**    In general, we adopt $\mathrm{Acc@}k\mathrm{IoU}$ as the evaluation metric following the standard setting of 3D visual grounding, where IoU is the Intersection over Union between the predicted 3D bounding box and the ground-truth bounding box formatted as $[center_x, center_y, center_z, size_x, size_y, size_z]$. To handle geometric ambiguities at multi-level visual grounding, we apply the following Eq. (1) for IoU computation. In our main paper, we set the threshold $t = 0.05(m)$, $k = 0.25$ and report $\mathrm{Acc@0.25IoU}$. For evaluations under other $k$ thresholds and explanation of the IoU formulation, please refer to the Appendix Section B.1.

$$
\mathrm{IoU} = \begin{cases} \mathrm{IoU}_{xy}^{2D}, & \text{if level = "area"} \\ \mathrm{IoU}_{\backslash i}^{2D} \cdot \mathbf{1}_{\left\{|\mathrm{center}_i^{\mathrm{gt}} - \mathrm{center}_i^{\mathrm{pred}}| < t \,\wedge\, \mathrm{size}_i^{\mathrm{pred}} < t\right\}}, & \text{if level} \neq \text{"area",} \\ & \mathrm{size}_i^{\mathrm{gt}} < t, i \in \{x, y, z\} \\ \mathrm{IoU}^{3D}, & \text{otherwise} \end{cases}
$$

(1)

**Baselines**    We evaluate three different types of models on our benchmark:

- **LLMs:** For each expression, the textual scene representation of LLMs are formatted as a scene graph, consisting of ground-truth locations and sizes of objects, as well as object captions. The object captions are generated using Qwen2.5-VL-72B [3] conditioned on multiple object images and a guided captioning instruction (see Appendix Section B.3 for comprehensive descriptions). Closed-source models, including non-thinking model GPT-4.1 [38] and thinking model o4-mini [39], and open-source models, including non-thinking (Qwen2.5 [49, 3]) and thinking models (Qwen3 [41], DeepSeek-R1-671B [20]), are benchmarked.

- **MLLMs:** Following the setting in GPT4Scene [40], we incorporate a bird's-eye view (BEV) image and eight uniformly sampled video frames with object markers as visual inputs, in addition to the textual scene representation used in the LLM setting. Closed-source models, including GPT-4.1 and o4-mini, as well as open-source models, including LLaVA-OneVision [31], Qwen2.5-VL [3], and GPT4Scene [40] are evaluated.

- **3D Visual Grounding Models:** We also evaluate four state-of-the-art specialized 3D visual grounding models: 3D-VisTA [64], PQ3D [65], Chat-Scene [24] and Grounded 3D-LLM [11]. Since Chat-Scene and Grounded 3D-LLM do not provide 3D features for datasets other than ScanNet, their evaluations are limited to the ScanNet portion of our benchmark.

Thorough experimental settings and implementations of baselines can be founded in Appendix Section B.2.

**Human Evaluation**    We construct a human evaluation subset of 200 expressions through stratified random sampling across four levels to maintain their original distribution. Human evaluators are instructed to annotate the corresponding 3D bounding boxes for each expression. Evaluation results on this subset are reported using the same metric as mentioned above.

## 3.2 Main Results

Table 2: Results are presented in Acc@0.25IoU on Anywhere3D-Bench. *object bbox* in the table denotes the ground-truth object locations and sizes for simplicity. Chat-Scene*, Grounded 3D-LLM*: evaluated only on ScanNet. Human**: performance evaluated on a subset of 200 expressions obtained through stratified random sampling across four levels.

| | Open Source | Area Level | Space Level | Object Level | Part Level | Overall |
|---|---|---|---|---|---|---|
| **LLMs:** *object bbox, captions* | | | | | | |
| **non-thinking** | | | | | | |
| GPT-4.1 | ✗ | 76.19 ± 0.75 | 17.28 ± 0.70 | 48.00 ± 0.45 | 22.94 ± 0.66 | 32.34 ± 0.08 |
| Qwen2.5-72B | ✓ | 60.14 ± 1.22 | 7.85 ± 0.30 | 33.30 ± 0.89 | 8.99 ± 1.50 | 19.90 ± 0.71 |
| Qwen2.5-VL-72B | ✓ | 56.35 ± 2.62 | 6.87 ± 0.35 | 29.19 ± 1.26 | 9.93 ± 1.86 | 18.05 ± 0.74 |
| **thinking** | | | | | | |
| o4-mini-2025-04-16 | ✗ | 71.96 ± 2.24 | 18.03 ± 0.23 | 48.69 ± 0.23 | 23.97 ± 0.53 | 32.80 ± 0.08 |
| Qwen3-32B(thinking) | ✓ | 59.79 ± 3.70 | 12.57 ± 0.36 | 40.18 ± 0.48 | 16.48 ± 1.04 | 25.51 ± 0.40 |
| DeepSeek-R1-671B-2025-01-28 | ✓ | 71.96 ± 1.40 | 14.61 ± 0.75 | 47.76 ± 0.12 | 20.92 ± 0.76 | 30.49 ± 0.48 |
| **MLLMs:** *object bbox, captions, BEV, video frames* | | | | | | |
| **non-thinking** | | | | | | |
| GPT-4.1-2025-04-14 | ✗ | 81.48 ± 2.25 | 19.03 ± 0.58 | 53.88 ± 1.04 | 25.85 ± 0.53 | 35.90 ± 0.34 |
| LLaVA-OneVision-7B | ✓ | 19.58 | 2.32 | 8.81 | 4.12 | 5.93 |
| Qwen2.5-VL-72B | ✓ | 57.16 ± 0.50 | 10.56 ± 0.83 | 40.74 ± 0.34 | 13.80 ± 1.32 | 24.19 ± 0.61 |
| GPT4Scene | ✓ | 15.34 | 7.19 | 25.16 | 11.99 | 14.55 |
| **thinking** | | | | | | |
| o4-mini-2025-04-16 | ✗ | 76.19 ± 2.24 | 23.00 ± 0.82 | 55.82 ± 1.41 | 31.46 ± 0.27 | 38.90 ± 0.10 |
| **3D visual grounding models:** *point clouds, video frames* | | | | | | |
| PQ3D | ✓ | 30.69 ± 0.92 | 8.36 ± 0.38 | 24.42 ± 0.18 | 16.73 ± 0.78 | 16.68 ± 0.04 |
| 3D-VisTA | ✓ | 29.10 ± 0.92 | 7.44 ± 0.38 | 25.05 ± 0.46 | 15.98 ± 0.28 | 16.26 ± 0.05 |
| Chat-Scene* | ✓ | 49.10 ± 2.70 | 6.55 ± 0.47 | 31.73 ± 0.31 | 22.99 ± 0.47 | 22.90 ± 0.37 |
| Grounded 3D-LLM* | ✓ | 49.25 | 6.62 | 26.36 | 19.37 | 20.10 |
| Human** | – | 100.00 | 92.00 | 98.00 | 97.00 | 95.00 |

Table 2 presents the overall results on our benchmark. Human performance substantially surpasses that of the best-performing model, o4-mini under the MLLM setting, particularly at the space level, indicating that current models fall far short of human-level 3D spatial intelligence.

**Area v.s. Space v.s. Object v.s. Part** Grounding expressions at the space level is the most challenging task on our benchmark. This difficulty arises from the need to understand spatial relations and distance, situations, and reason over the absolute locations in 3D space beyond objects. The best-performing model, o4-mini with visual reasoning ability, only achieves 23.00% accuracy. Part-level visual grounding, though derived from object-level grounding, also poses significant challenges for all models. It requires the model to first identify the object to which the part belongs, and then reason about the part's location and size based on spatial relationships, functions, and other contextual cues. In contrast, area-level and object-level grounding are relatively easier.

**LLMs v.s. MLLMs** Additional visual inputs, i.e., video frames and the bird's-eye view image, consistently improve the performance of the same models (GPT-4.1, o4-mini, and Qwen2.5-VL-72B) when transitioning from the LLM setting to the MLLM setting. The performance gains are notable at object level (8.19% on average) and part level (4.76% on average), as models can leverage visual inputs to access richer information about the object's details, such as color and structure. However, improvements at the space level are less pronounced (3.47% on average), suggesting that current MLLMs have limited ability to interpret spatial relationships in 3D space from 2D images.

**MLLMs v.s. 3D Visual Grounding Models** Both MLLMs and 3D visual grounding models are provided with visual inputs, along with the ground-truth bounding box locations and sizes. However, specialized 3D visual grounding models demonstrate limited performance, particularly at the space level, as they are restricted to predicting objects in the scene and lack generalizability for multi-level visual grounding tasks.

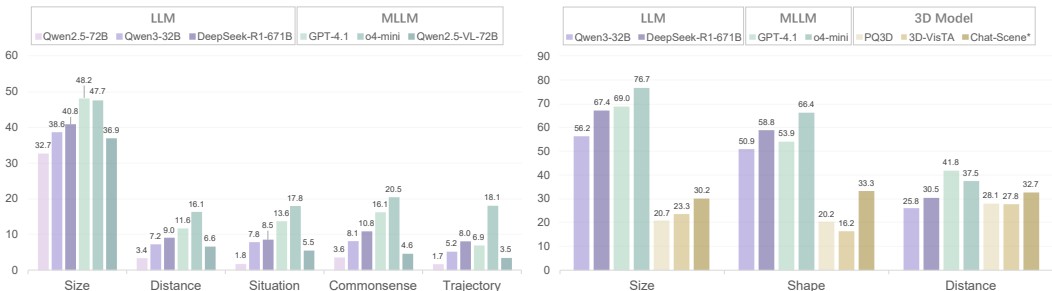

Figure 4: Results on different types of expressions on Space Level.

Figure 5: Results on different types of expressions on Object Level.

## 3.3 Detailed Analysis on Grounding Levels

Furthermore, we examine model performance across different types of referring expressions at each visual grounding level. Due to space limitations, we leave the analysis on area-level visual grounding to the Appendix Section B.7.

**Space Level** We report the top-performing LLMs and MLLMs on different types of expressions at the space level, as illustrated in Fig. 4. Trajectory-based expressions are comparatively challenging, as they require a comprehensive understanding of spatial distance, relationships, and orientation. Expressions involving situation, distance, and commonsense also present difficulties, as they demand reasoning about spatial regions beyond objects in the scene. In contrast, size-related expressions are relatively easier: selecting the correct object, adjusting its size based on instructions, and performing positional refinements require less complex spatial perception and reasoning.

**Object Level** Fig. 5 demonstrates the performance of three types of expressions at object level. An interesting observation is that, compared to MLLMs, 3D visual grounding models exhibit a more balanced capability in interpreting object size, shape, and distances between objects. This may be attributed to the fact that 3D models are typically trained on point clouds, which inherently encode spatial coordinates. For MLLMs, object sizes are explicitly provided in the scene graph, contributing to their stronger performance on size-related tasks, whereas estimating distances between objects requires both complex computation and commonsense reasoning.

Despite 3D visual grounding models' relatively balanced ability to understand size and distance on the object-level tasks, they still underperform on our benchmark overall. Compared to visual grounding results reported on ScanRefer [7], where state-of-the-art 3D models achieve around 50% accuracy, these models' performance show a substantial performance drop on our benchmark. This suggests that reasoning about quantitative object size and inter-object distance remains a significant challenge for current 3D visual grounding approaches.

**Part Level** Fig. 7 shows the performance of top-performing models on different types of expressions at part level. Expressions involving dynamic movement present the greatest challenge for all models, as models must not only accurately identify the specific part of the object but also understand the object's orientation to correctly predict the position of the bounding box after the movement. For expressions involving spatial relationships, the best-performing non-thinking model (GPT-4.1) performs worse than the 3D visual grounding model (Chat-Scene*), while the thinking model (o4-mini) achieves only a small margin higher performance than Chat-Scene*, indicating that MLLMs still struggle with spatial relationships in 3D environments.

## 4 How Can We Improve MLLMs' Ability on Multi-level Visual Grounding?

The above experimental results and analysis indicate that space-level and part-level visual grounding are the most challenging tasks in our benchmark. We select several representative examples from the second-best-performing model GPT-4.1, as the best-performing model o4-mini does not provide a reasoning process for detailed analysis. As illustrated in Fig. 6(a), (b), and (c), on space-level tasks, GPT-4.1 struggles with expressions involving situated contexts and trajectory, all of which

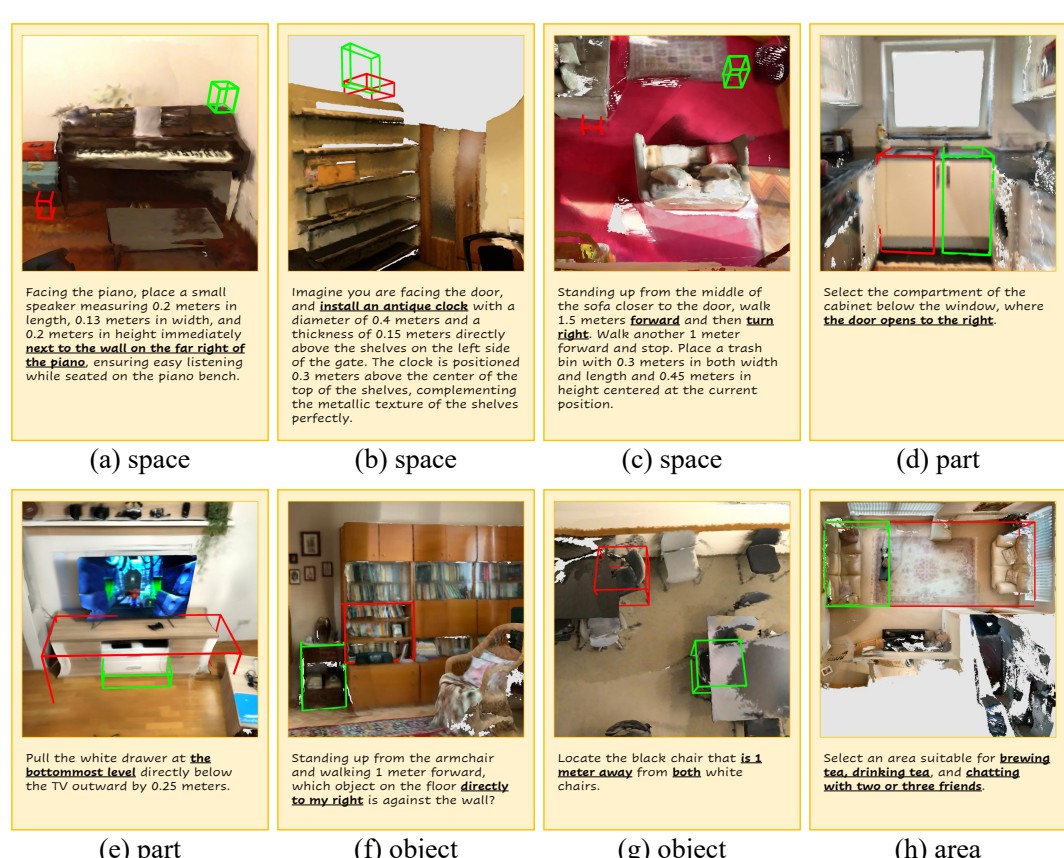

Figure 6: Qualitative examples from Anywhere3D-Bench. Green boxes indicate ground-truth, while red boxes show predictions from GPT-4.1. Examples (a)–(c): space-level; Examples(d) and (e): part-level; Examples (f) and (g): object-level; Example (h): area-level.

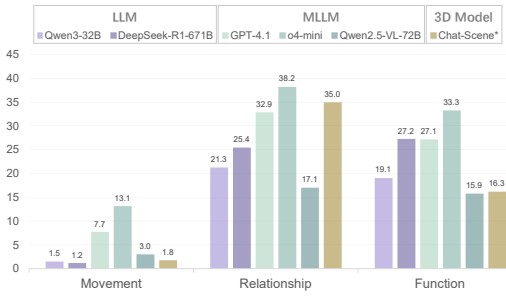

Figure 7: Results on different types of expressions on Part Level.

Table 3: Effect of the visual perception enhancement and the relational reasoning enhancement separately. Δ denotes the change in accuracy relative to GPT-4.1. The reported results are based on the human evaluation subset and averaged across three independent trials.

| Method | Area | Space | Object | Part | Overall |
|---|---|---|---|---|---|
| *GPT-4.1* | 86.67 | 15.29 | 54.29 | 33.33 | 37.00 |
| Δ(GPT-keyframe) | 4.44 ↓ | 2.75 ↑ | 2.86 ↓ | 8.90 ↑ | 1.00 ↑ |
| Δ(Human-keyframe) | 6.66 ↑ | 7.85 ↑ | 3.33 ↑ | 15.57 ↑ | 7.50 ↑ |
| Δ(BEV-axes) | 6.66 ↑ | 1.96 ↑ | 2.85 ↑ | 5.56 ↑ | 3.00 ↑ |
| Δ(BEV-axes + ori.) | 2.22 ↑ | 5.49 ↑ | 0.96 ↓ | 3.34 ↑ | 4.00 ↑ |

require relational reasoning over object orientations, spatial directions, and spatial locations. On part-level tasks, as illustrated in Fig. 6(d) and (e), GPT-4.1 encounters difficulties in identifying specific parts from objects, due to the uniformly sampled video frames being insufficient to capture the key visual cues necessary for resolving the referring expressions. To address these challenges, we introduce visual and relational input enhancements as initial attempts to improve model performance, as detailed in the following section.

## 4.1 Visual Perception Enhancement

To provide the model with more query-relevant visual cues, we incorporate key video frames **related to** the referring expressions in addition to the original visual inputs (i.e. **uniformly sampled** video frames and bird's-eye view image). Specifically, given the original textual and visual inputs, we first prompt GPT-4.1 to output the candidate object IDs that it considers relevant to the referring expression. For each candidate object, we then select one video frame in which the object appears with the largest relative size and highlight it with a green 2D bounding box. Incorporating all these selected video frames alongside the original inputs, we then require GPT-4.1 to predict the 3D bounding box corresponding to the expression.

Moreover, to illustrate the "upper bound" performance of incorporating additional selected video frames, we manually annotate the object IDs that humans consider relevant to each referring expression and select video frames accordingly. These video frames, combined with the original inputs, are then used as input to GPT-4.1.

As demonstrated in Table 3, incorporating additional related video frames improves GPT-4.1's performance on *Anywhere3D-Bench*, particularly at the part level, which demands finer-grained object details. However, a performance gap remains between the two object selection settings, i.e., objects proposed by the GPT-4.1 and those identified by humans, indicating GPT-4.1's limitations in accurately selecting relevant objects.

## 4.2 Relational Reasoning Enhancement

To provide more spatial relation information to the MLLM, we adopt two approaches: scene-level enhancement and object-level enhancement. For scene-level enhancement, we overlay the x-axis and y-axis on the bird's-eye view (BEV) image of the scene. For object-level enhancement, we incorporate object orientations—predicted by Orient-Anything [45]—into the textual scene graph input. To ensure consistency with the scene graph coordinate system, object orientations are discretized into +x, -x, +y, -y, or "not sure," according to predictions from Orient-Anything.

As illustrated in Table 3, the combined use of scene-level and object-level enhancements improve GPT-4.1's performance at the space level and part level. However, the accuracy remains far behind human performance. This highlights a substantial gap in spatial reasoning ability, even when models are provided with enriched spatial cues.

For comprehensive demonstrations on qualitative examples in Fig. 6, as well as the implementations of visual perception enhancement and relational reasoning enhancement, please refer to Appendix Section C.

## 5 Related Work

**3D Visual Grounding** 3D vision-language learning establishes critical connections between natural language and 3D environments, enabling applications in augmented/virtual reality [8, 60] and embodied AI systems [16]. 3D visual grounding—the precise localization of language-referred entities in 3D scenes—has emerged as a cornerstone for spatial intelligence. Despite the proliferation of benchmarks [7, 1, 58, 30, 63, 26], existing datasets remain predominantly object-centric, constraining models to coarse-grained scene understanding. Recent efforts like SceneFun3D [15] partially address this limitation by introducing a predefined set constrained on small functional elements (e.g., handles, buttons). In contrast, Anywhere3D involves more open-ended object parts (e.g., toilet tank, lampshade of the lamp) and emphasizes the visual grounding of part movements, as shown in Fig. 1. MMScan [35] introduces region-level visual grounding, which extends object-centric tasks to human-activities regions, similar to our area-level tasks. However, it does not involve visual grounding at **unoccupied space**, such as placing a new object or moving an existing object to a specified unoccupied space within the scene. Concurrently, while advanced visual grounding methods [64, 65, 12, 21, 46, 34, 28, 62, 9, 52, 42, 47, 33, 59, 5, 50, 55, 66] demonstrate progress in object-level localization, their capacity to interpret referrals at multi-levels remains underexplored. Our benchmark bridges this gap by introducing multi-granular localization across four hierarchical levels— *area, space, object*, and *part*—systematically evaluating model performance in complex, real-world 3D scene grounding.

**Evaluating MLLMs on 3D Spatial Understanding**    Recent advancements in LLMs have facilitated their integration into 3D domains. Early approaches, often termed "3D LLMs," such as 3D-LLM, LEO, and Chat-Scene [22, 48, 32, 19, 23, 10, 25, 24], fine-tune LLMs to process embedded 3D object features. However, fine-tuning for 3D tasks is computationally expensive and risks catastrophic forgetting [56]. In contrast, GPT4Scene [40] demonstrates that MLLMs can effectively tackle 3D understanding through simple visual prompting, bypassing the need for task-specific adaptation, which highlights the untapped potential of MLLMs in 3D intelligence. Concurrently, there is a growing interest in benchmarking off-the-shelf MLLMs on 3D tasks. VSI-Bench [51] evaluates 3D spatial reasoning in video understanding, while All-Angles Bench [53] tests MLLMs' ability to establish correspondence between multi-view visual data. ScanReQA [57] further investigates how multimodal inputs affect spatial reasoning, comparing traditional 3D LLMs and MLLMs. Space3D-Bench [43] encompasses a variety of spatial tasks—including object localization, spatial measurements, and navigation—that span both objects and entire rooms. Despite these efforts, the field has yet to systematically assess the 3D visual grounding abilities of MLLMs, leaving open questions about their precision in localizing and reasoning within complex spatial scenes. For detailed discussion and comparison of these works, please refer to Appendix Section D.

## 6    Conclusion

In this paper, we present *Anywhere3D-Bench*, a novel and challenging benchmark that extends visual grounding to four levels in 3D scenes. Evaluation results show that even the best-performing MLLM, o4-mini, with visual reasoning capabilities, struggles with the two most difficult tasks—space-level and part-level grounding. This highlights the difficulty current MLLMs face in understanding and reasoning about 3D scenes based on 2D visual inputs. Furthermore, specialized 3D visual grounding models consistently underperform compared to MLLMs, particularly on space-level tasks, revealing their limited generalizability to multi-level grounding scenarios.

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

## A    Anywhere3D Benchmark

### A.1    Data Statistics

We first present the number of referring expressions across the four grounding levels on ScanNet, MultiScan, 3RScan, and ARKitScenes, as shown in Table A1. To visually demonstrate the linguistic diversity of referring expressions in *Anywhere3D-Bench*, we generate a word cloud based on all expressions, as illustrated in Fig. A1.

Table A1: Number of referring expressions per grounding level across ScanNet, Multiscan, 3RScan and ARKitScenes.

| Dataset | Area Level | Space Level | Object Level | Part Level |
|---|---|---|---|---|
| ScanNet | 93 | 498 | 643 | 245 |
| MultiScan | 5 | 56 | 17 | 20 |
| 3RScan | 16 | 197 | 92 | 67 |
| ARKitScenes | 75 | 458 | 202 | 202 |

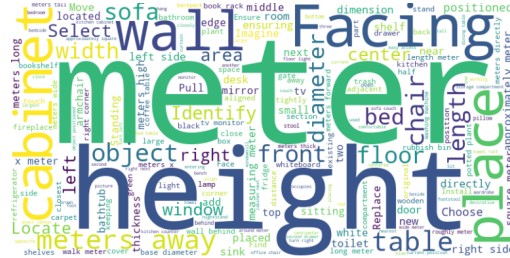

Figure A1: Word cloud of *Anywhere3D-Bench*

Furthermore, we conduct a distributional analysis of object-level expressions with respect to object size, floor area, and inter-object distance. The results reveal a broad spectrum of referents, ranging from small to large objects and from proximate to distant spatial references, underscoring the diversity of expressions captured in our benchmark (see Fig. A2, Fig. A3, and Fig. A4).

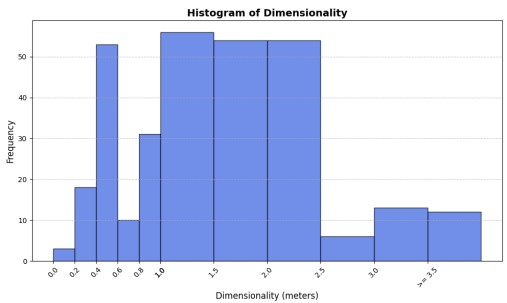

Figure A2: Dimensionality Distribution of referring expressions at object level.

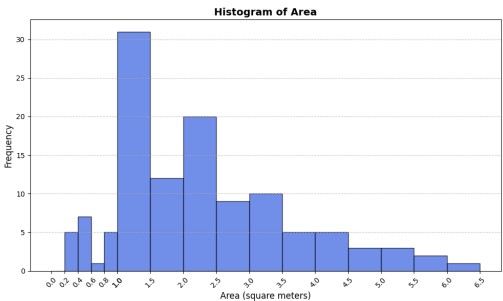

Figure A3: Floor Area Distribution of referring expressions at object level

To ensure that models predict answer based on understanding and reasoning on object size and inter-object distance, rather than simply matching object labels in the referring expressions, we explicitly exclude ground-truth object labels from the expressions at the object level, unless there are multiple instances of the target object in the scene. This design encourages models to identify the correct object based on 3D scene understanding rather than relying on linguistic cues tied to object categories. As a result, 50% of the referring expressions do not contain the corresponding ground-truth object label, as they refer to objects with unique instance in the scene. The remaining expressions refer to objects with multiple instances, as illustrated in the distribution shown in Fig. A5.

As shown in Fig. A6, the volume distribution of Anywhere3D targets exhibits greater dispersion and heavier tails on both ends, indicating more frequent extreme small and large volumes compared to ScanRefer, where volumes are tightly concentrated around the mean. This difference arises because Anywhere3D expands grounding granularity beyond object-level annotations to include spaces, *parts* (typically small), and *areas* (typically large), enabling a more comprehensive evaluation of 3D visual grounding.

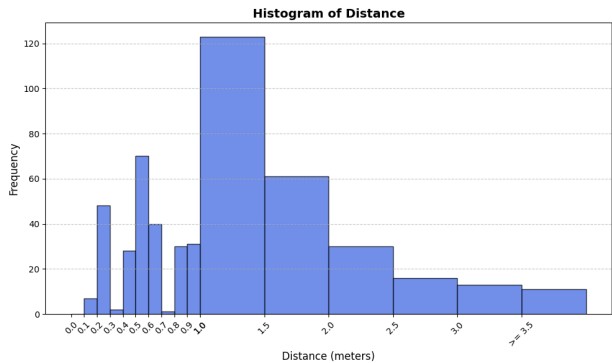

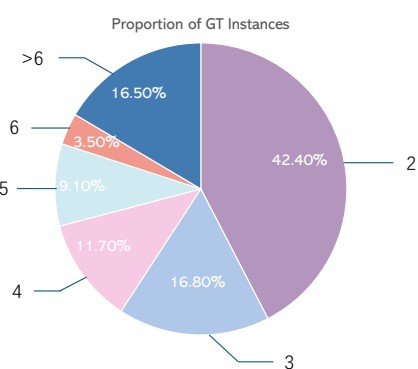

Figure A4: Distance Distribution of referring expressions at object level.

Figure A5: Distribution of object categories with two or more ground-truth instances at the object level. Only in these cases is the object label contained in the referring expression.

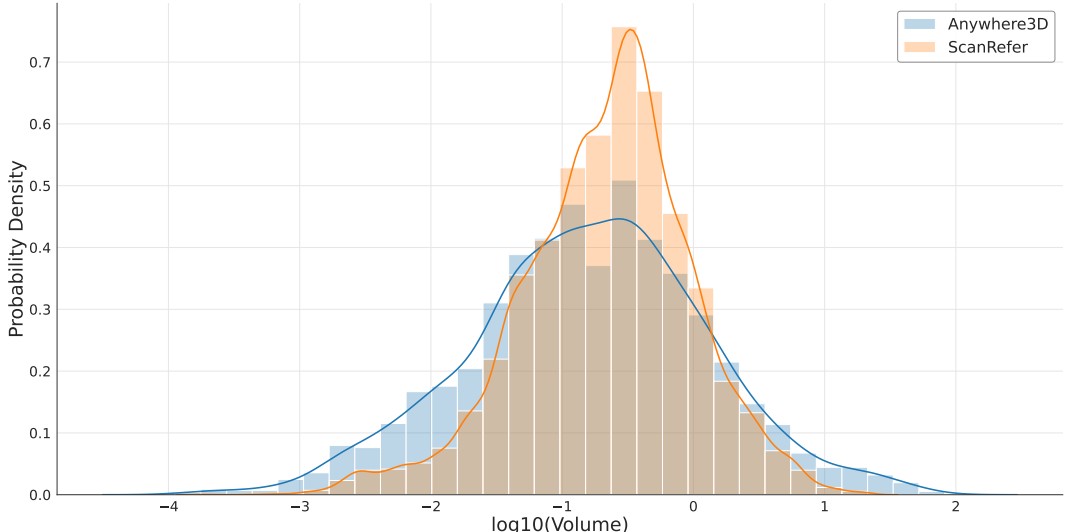

Figure A6: The target objects' volume distribution of Anywhere3D and ScanRefer. Logarithmic scaling is applied to the horizontal axis.

## A.2 Data Generation Details

As demonstrated in Fig. A7, Fig. A8, and Fig. A9, we present the prompt messages for **distance-related** referring expressions generation at **space level** as an example.

> messages = [{'role': 'system', 'content': System prompt}, {'role': 'user', 'content': Scene graph of the scene to process}]

Figure A7: Prompt messages for referring expression generation with GPT-4o

## A.3 Human Annotation and Verification Details

### A.3.1 Annotation Interface

Fig. A10 and Fig. A11 illustrate the human annotation interface, which is adapted from ScanRefer. Our interface comprises four main components: a control bar, a 3D scene visualization module, an object list, and a referring expression editing box.

You are now a helpful assistant that can generate diverse referring expressions that can be grounded to reasonable space in an indoor scene.

The scene is represented by a scene graph in JSON dictionary format. Each entity in the scene graph denotes an object instance, named '<category>-<ID>'. The 'caption' field describes the object's attributes, such as 'color', 'material', etc. The 'relations' field specifies the object's spatial relationships with other objects, defined from a viewpoint along the y-axis from positive to negative direction. The 'position' field contains the x, y, z coordinates of object's center in the scene. The 'size' field describes the width, length and height of the object's 3D bounding box. The numerical values of 'position' and 'size' correspond to units in meters. For example, from the scene graph:

'''
"object_info": "kitchen counter-1": "relations": ["below cabinet-4", "lower than soap dispenser-14", "lower than paper towel dispenser-15"], "position": [1.27, 0.67, 0.9], "size": [0.74, 1.86, 0.26], "caption": "The kitchen counter is black granite with a stainless steel sink and faucet ... It is durable, easy to clean, and has a modern, sleek design that matches the stainless steel appliances in the kitchen.", ...
'''

You can know that the center of "kitchen cabinets-5" is located in the x: 1.01, y: 0.37, z: 0.45, the "floor-6" has the width and length of 4.81 meters and 3.05 meters, the "microwave-8" is placed within the area of the "cabinets-3", the "cabinet-2" is to the left of "water cooler-7" viewing from +y axis to -y axis, the "water cooler-7" is 3 o'clock direction near the "cabinet-2" viewing from +y axis to -y axis.

Using the provided scene graph, design referring expressions that can be grounded to reasonable space in the 3D scene. There are two principles you need to read and follow very carefully:

1. Clarity: Each Referring Expression must be grounded exactly to one target 3D bounding box in space. Do not include the IDs of the objects in the referring expressions. Instead, use ordinal words, colors and relations to refer to different object instances of the same category. Describe the grounding position of the target 3D bounding box using only the surrounding objects to avoid causing confusion. Avoid using terms like "o' clock" to describe relations in referring expressions. Do not refer to existing objects with their corresponding positions in the scene! Additionally, consider whether the expressions require a specific viewpoint to ensure the target bounding box is clearly and uniquely identifiable. In some cases, specifying a viewpoint is necessary for achieving this level of precision. Please note that you don't need to stick to the original viewpoint in the scene graph (which is along the y-axis, from the positive to the negative), but if you specify a viewpoint, the spatial relations between objects in your referring expressions need to be consistent with the scene.

2. Distance Understanding Related: You should generate referring expressions in which the position of the bounding box should be explicitly specified based on its numerical distance from other objects in the scene. There are two main categories of referring expressions: existing object movement (move object already existed in the scene to another place) and new object increment (Add objects that not exist in the scene). For the category of 'new object increment', you may assume objects that not exist in the scene graph in each of the referring expressions. However, they should have reasonable sizes(For example, they should not exceed the boundary of the scene, not overlap with the existing objects.) and should be placed reasonably in the scene (For example, they are not allowed floating in the air). Also, you need to explicitly provide the sizes, i.e. (WIDTH, LENGTH, HEIGHT) or (DIAMETER, HEIGHT). You can also generate referring expressions beyond these categories that meet the principles above.

Below are some example referring expressions. Please note that these examples are derived from different scene graphs.

EXAMPLES

After you understand the contents above, I will provide a new scene graph below. Based on the two guiding principles and the examples provided above, generate referring expressions corresponding to the new scene graph.

Figure A8: System Prompts for generating distance-related referring expressions at the space Level using GPT-4o

1. Move the smaller trash bin next to the refrigerator to a spot 0.9 meters directly in front of the bicycle.

2. Facing the shower curtain, move the rug on the floor 0.3 meters to the right.

3. Pull the piano bench out by 0.5 meters to allow space for someone to sit and play the piano.

4. Move the trash can that is closest to the TV to the floor directly in front of the sink, placing it 0.7 meters away from the kitchen cabinet.

5. Shift the television on the wall 0.1 meters to the right.

6. Facing the tv, move the coffee table 0.55 meters to its left.

7. Place a basin with a radius of 0.25 meters and a height of 0.5 meters on the floor, centered 0.35 meters to the right of the sink.

8. Place a round coffee table 0.8 meter in front of the couch. The table has a diameter of 1.2 meter and a height of 0.7 meter.

9. Move the painting on the wall next to the bed downward so that the bottom of the painting is 1 meter above the floor.

10. Add a new table in front of the big sofa with dimensions 80 cm × 60 cm × 1 m. The table should be placed 1.5 meters away from the sofa and centered relative to it.

11. Sitting on the chair in front of the desk, place a notebook with 0.4m * 0.5m * 0.1m to the right of the laptop. Keep a distance of 0.15 meters between them.

12. Facing the whiteboard, add another foosball table in font of the old one, remaining 0.2 meters between the two tables.

Figure A9: Examples in System Prompt for GPT-4o referring expressions generations in distance-related expressions at space level.

The control bar, located in the top-left corner, includes three primary tools: (1) a 3D bounding box annotation tool, (2) a distance measurement tool (i.e. *Scale Cylinder*), and (3) a coordinate axis visualization tool. Both the bounding box and distance measurement cylinder can be resized and repositioned using the control bar, and can also be interactively adjusted with the mouse. The dimensions 'W', 'L', and 'H' represent the lengths of the bounding box along the x-, y-, and z-axes, respectively.

The 3D scene visualization module supports interactive operations such as zooming and rotating, allowing annotators to conduct detailed spatial exploration in the scene.

The object list, located in the top-right corner, displays all objects in the scene, along with their associated labels and sizes.

The referring expression editing box presents the expressions initially generated by GPT-4o and provides an interactive field for manual revision. Annotators can save their annotations or load previously saved ones as needed.

### A.3.2 Statistics on final referring expressions after human verification

Here, we provide quantitative statistics on the divergence of final expressions after human verification from the original expressions generated by GPT.

Before annotation began, we established clear guidelines: human annotators are allowed to revise the GPT-4o-generated expressions, but they first had to estimate the proportion of modification within the expression. If the required changes exceeded 50% of the original expression, annotators were allowed to "skip" that referring expression. Expressions needing such extensive revision were filtered out.

Overall, 25% of the candidate expressions were marked as "skip" and thus excluded from the dataset. For the remaining expressions, the **Average Modification Ratio** is approximately 42%, as calculated by Eq. (A1):

$$\text{Average Modification Ratio} = \frac{1}{N} \sum_{i=1}^{N} \frac{\text{Levenshtein}\left(E_i^{\text{GPT}}, E_i^{\text{Human}}\right)}{|E_i^{\text{GPT}}|} \tag{A1}$$

where $N$ denotes the total number of referring expressions, Levenshtein($E_i^{\text{GPT}}, E_i^{\text{Human}}$) represents the Levenshtein Distance (i.e., the minimum number of single-word edits required to transform the GPT-4o-generated expression $E_i^{\text{GPT}}$ into the final human-verified expression $E_i^{\text{Human}}$), and $|E_i^{\text{GPT}}|$ is the length of the original GPT-4o-generated expression.

### A.3.3 Total Cost and Duration of the Human Annotation & Verification Process

Overall, human annotation and verification process for all referring expressions cost approximately 900 USD and took around six weeks, including time for tool familiarization, pilot annotation, formal annotation, and human verification.

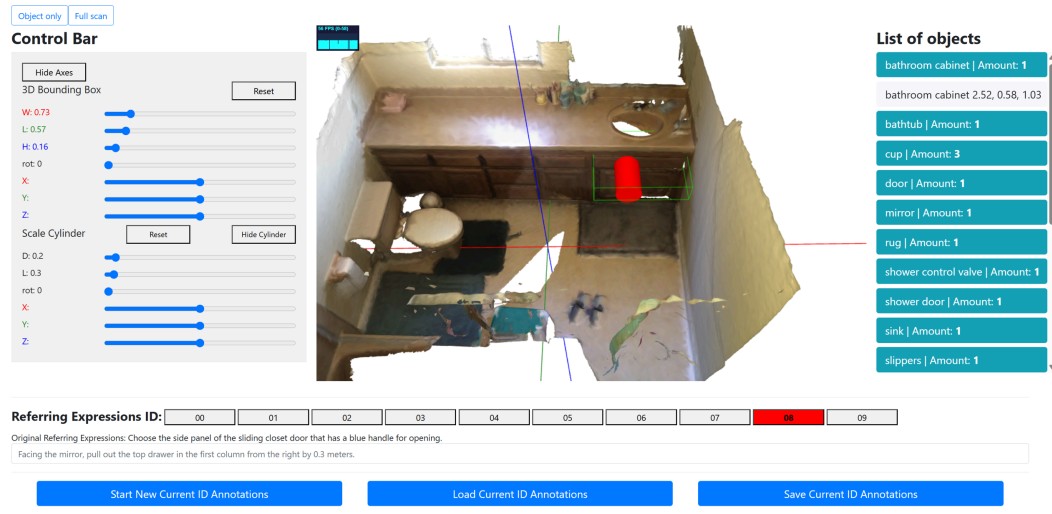

Figure A10: Human Annotation at part level: "Facing the mirror, pull out the top drawer in the first column from the right by 0.3 meters.".

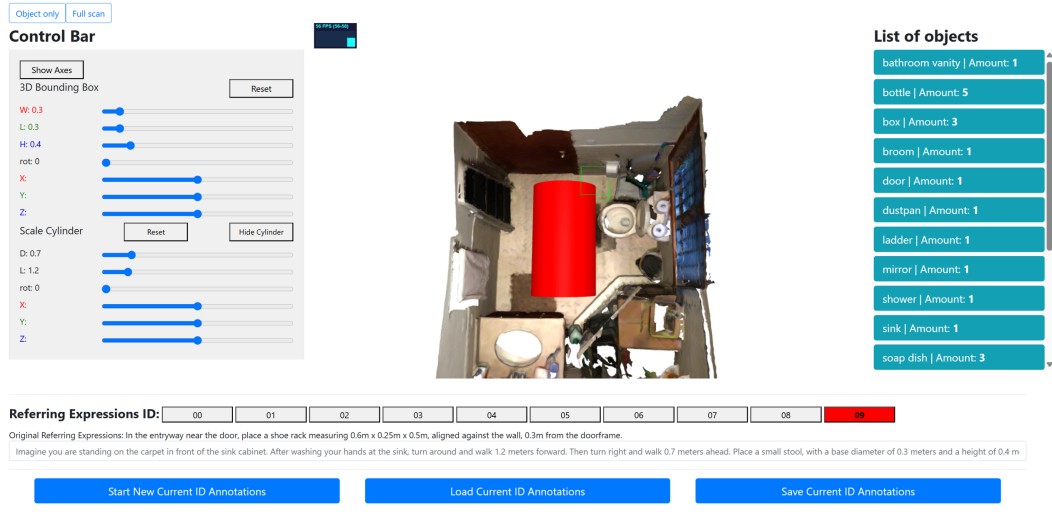

Figure A11: Human Annotation at space level: "Imagine you are standing on the carpet in front of the sink cabinet. After washing your hands at the sink, turn around and walk 1.2 meters forward. Then turn right and walk 0.7 meters ahead. Place a small stool, with a base diameter of 0.3 meters and a height of 0.4 meters, centered at your current position." The scale cylinder serves as good annotation tool for trajectory annotation at space level.

# B Experiments and Results

## B.1 Evaluation Metrics

In this section, we provide a detailed explanation of the IoU computation formula, as shown in Eq. (B1).

$$\text{IoU} = \begin{cases} \text{IoU}^{2D}_{xy}, & \text{if level} = \text{``area''} \\ \text{IoU}^{2D}_{\setminus i} \cdot \mathbf{1}_{\left\{|\text{center}^{\text{gt}}_i - \text{center}^{\text{pred}}_i| < t \,\wedge\, \text{size}^{\text{pred}}_i < t\right\}}, & \text{if level} \neq \text{``area''}, \\ & \text{size}^{\text{gt}}_i < t, i \in \{x, y, z\} \\ \text{IoU}^{3D}, & \text{otherwise} \end{cases}$$

$$(\text{B1})$$

For the area level, we adopt 2DIoU on the horizontal plane as the evaluation metric, due to the inherent ambiguity in defining the vertical extent of the bounding box. For instance, it is unclear whether a study area comprising a desk and an office chair should extend vertically to the ceiling, as illustrated in Fig. B1.

Moreover, due to inevitable rendering artifacts in mesh visualization during human annotation, there can be small positional deviations—typically within a few centimeters. For certain objects with very small dimensions along one axis (e.g., a floor carpet that is only 0.02 meters thick(Fig. B2), or a wall clock with a thickness of 0.03 meters), directly computing the 3D IoU can lead to significantly distorted results, as the metric becomes overly sensitive to minor positional misalignments. To address this, we design a customized evaluation strategy for such objects.

Specifically, given a predefined threshold, if the ground-truth bounding box has a dimension smaller than this threshold along any axis (x, y, or z), and if the predicted bounding box satisfy both following conditions: (1) deviates from the ground-truth bounding box by less than the threshold along that axis in terms of position, and (2) is also smaller than the threshold in size along that axis, we consider the prediction to be accurate along that axis in both position and size. In this case, the IoU is computed as a 2D IoU over the remaining two axes only. We set the value of threshold to 0.05 meters in the paper.

Table B1 illustrates the ground-truth 3D bounding box and predicted 3D bounding box corresponding to Fig. B2. The height of the carpet(i.e., the size in the z-axis) is smaller than 0.05 meters. The predicted bounding box satisfies both of the conditions above, so the IoU between the predicted bounding box and the ground-truth bounding box will be the 2DIoU in the XY-Plane.

Table B1: 3D bounding box of ground-truth and prediction of GPT-4.1 corresponding to Fig. B2. The predicted 3D bounding box meets the criteria mentioned above, so IoU between ground-truth and prediction will be 2D IoU on the x-y plane $\approx \mathbf{0.4696}$

|  | center_x | center_y | center_z | size_x | size_y | size_z |
|---|---|---|---|---|---|---|
| ground-truth | -0.01 | -1.03 | -0.03 | 1.8 | 1.2 | 0.02 |
| prediction | -0.01 | -0.68 | 0.01 | 1.2 | 1.8 | 0.02 |

## B.2 Baseline Settings

To encourage diverse reasoning, we use the default temperature settings for all models, including GPT-4.1, o4-mini, Qwen3-32B, Qwen2.5-72B, Qwen2.5-VL-72B, DeepSeek-R1-671B, and DeepSeek-V3-671B. Each model is evaluated independently over three runs. The mean and standard deviation reported in the main paper and the Appendix are computed based on these runs. Table B2 illustrates the specific versions of the LLMs and MLLMs used.

Table B2: Versions of LLMs and MLLMs.

| Model | Qwen2.5-72B | Qwen3-32B | DeepSeek-R1-671B | DeepSeek-V3-671B | GPT-4.1 | o4-mini | Qwen2.5-VL-72B |
|---|---|---|---|---|---|---|---|
| Version | 2024-09-19 | 2025-04-29 | 2025-01-20 | 2024-12-26 | 2025-04-14 | 2025-04-16 | 2025-01-27 |

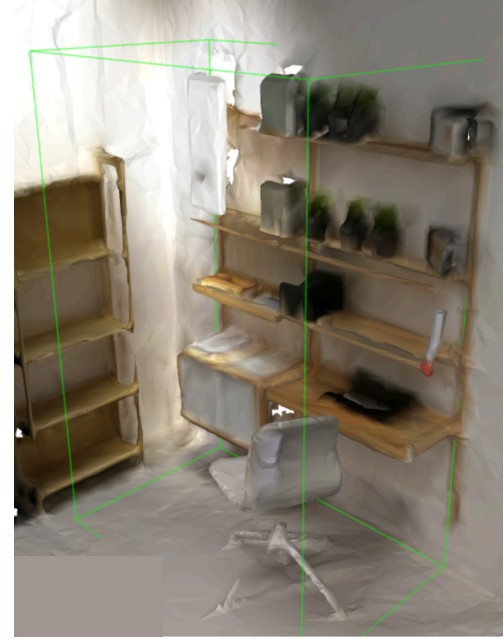 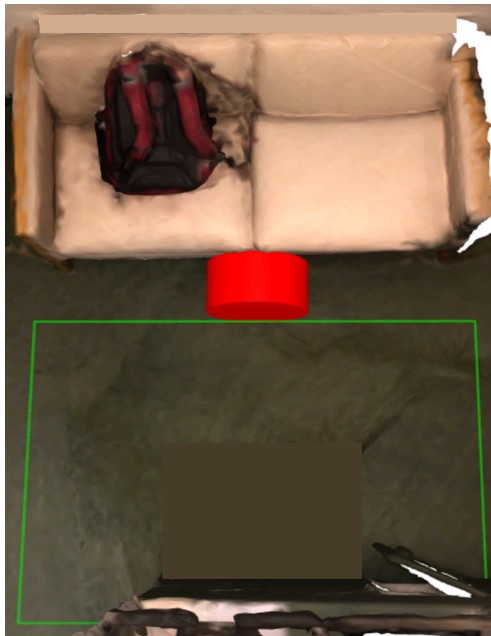

Figure B1: "Choose the area suitable for a study or home office setting."

Figure B2: "Lay a new rectangular carpet on the floor directly in front of the sofa, ensuring it is 0.2 meters away from the sofa. The carpet should measure 1.2 meters wide, 1.8 meters long, and **0.02 meters thick.**"

For both LLMs and MLLMs, we design system prompts tailored to each visual grounding level. The textual prompts consist of object ground-truth bounding boxes and captions generated by Qwen2.5-VL-72B (as further illustrated in Appendix Section B.3). In addition to textual inputs, MLLMs also receive a Bird's Eye View (BEV) image and eight uniformly sampled video frames as visual inputs, following the setup of GPT4Scene. Fig. B3 and Fig. B4 show the textual inputs for LLMs at **space level**, while Fig. B5 and Fig. B6 present the textual inputs for MLLMs at **space level**. An example of the BEV and video frames used as visual inputs for MLLMs is shown in Fig. B7.

messages = [{'role': 'system', 'content': System prompt for specific grounding level}, {'role': 'user', 'content': "Object_info": Scene graph of the scene + "\n" + "Referring_expression": referring expression}]

Figure B3: textual inputs for LLMs

Notably, we provide both LLMs and MLLMs with the ground-truth locations and sizes of objects. This design decouples object detection from 3D visual grounding, allowing us to specifically examine the models' capabilities in perceiving, understanding, and reasoning within 3D scenes under ideal localization conditions.

After obtaining the model's output, we extract the predicted 3D bounding box location and size using a combination of regular expressions and LLM-based parsing.

We utilize the fine-tuned checkpoints released by LLaVA-OneVision, GPT4Scene, PQ3D, 3D-VisTA, Chat-Scene, and Grounded 3D-LLM. Among them, GPT4Scene, PQ3D, 3D-VisTA, Chat-Scene, and Grounded 3D-LLM can only predict object IDs corresponding to the referring expressions. Therefore, we derive the predicted 3D bounding boxes based on the size and location of the corresponding objects in the scene graph. Moreover, Grounded 3D-LLM fails to produce visual grounding results on some scenes of ScanNet due to feature misalignment. As a result, our evaluation is restricted to those scenes where results can be successfully generated.

You are now a helpful assistant capable of grounding referring expressions to specific space within a 3D scene.

The scene is represented by a scene graph in JSON dictionary format. Each entity in the scene graph denotes an object instance, named '<object>-<ID>'. The 'position' field contains the x, y, z coordinates of object's 3D bounding box center in the scene. The 'size' field indicates the length of the object's 3D bounding box along the x, y, and z axes. Note that the x and y axes correspond to the horizontal plane, while the z axis corresponds to the vertical direction. The numerical values of 'position', 'size' correspond to units in meters. The "caption" field contains textual description of the object generated by a vision-language model (VLM) based on several images of the object. For example, from the scene graph:

""" 
"object_info": "object-1": "position": [1.27, 0.67, 0.9], "size": [0.74, 1.86, 0.26], "caption": "The kitchen counter is black granite with a stainless steel ...", ...
"""

You can know that the center of "object-5" is located in the x: 1.01, y: 0.37, z: 0.45, the "object-6" has the length and width of 4.81 meters and 3.05 meters.

"Referring expressions" are natural language descriptions that point to a specific space within a 3D scene, which is represented by a scene graph.

For example, a referring expression like "Facing the bed, move the nightstand 0.3 meters backward." requires identifying the 3D bounding box of the nightstand after it has been moved 0.3 meters backward.

Your task is to determine the 3D bounding box corresponding to the referring expression and return the following details:

1. The x, y, z coordinates of the center of the bounding box.

2. The lengths of the bounding box along the x, y, and z axes.

After reviewing the information above, I will provide a new scene graph and a referring expression. Your task is to identify the 3D bounding box that corresponds to the referring expression within the new scene graph.

At the end of your response, please provide the following details for the identified 3D bounding box:

1. The x, y, z coordinates of its center, formatted strictly as: {xcoordinate: , ycoordinate: , zcoordinate: }

2. The length of the 3D bounding box along the x, y, and z axes, formatted strictly as: {xlength: , ylength: , zlength: }

Figure B4: System Prompts of LLMs textual inputs for space level

messages = [{'role': 'system', 'content': System prompt for specific grounding level}, {'role': 'user', 'content': "Object_info": Scene graph of the scene + "\n" + "Referring_expression": referring expression + "\n" + "The subsequent images include a Bird Eye View image as the first, followed by 8 frames extracted from the scene video. Please return the center coordinates and sizes of predicted 3D bounding box STRICTLY following the instructed format."}]

Figure B5: textual inputs for MLLMs

## B.3 Object Caption Generation for Benchmarking

For LLMs and MLLMs, as outlined in Section B.2, the inputs include object captions generated by Qwen2.5-VL-72B. For each object in the scene, we prompt Qwen2.5-VL-72B to generate a descriptive caption using up to **five** *uniformly* sampled frames where the object is visible. To guide the captioning process, we annotate each frame with a green bounding box—projected from 3D space—to highlight the target object.

The prompt instructs Qwen to describe the object within the bounding box, covering its category, material, color, shape, structure, function, and surrounding environment. The full prompt template is provided in Fig. B9.

An example is illustrated in Fig. B8

You are a helpful assistant skilled in grounding referring expressions to specific space within a 3D scene.

Each scene is represented by the following elements:

1. scene graph: a JSON-formatted dictionary that enumerates all objects in the scene. Each entity in the scene graph denotes an object instance, named '<object>-<ID>'. For each object, the 'position' field contains the x, y, z coordinates of the center of its 3D bounding box. The 'size' field indicates the length of the 3D bounding box along the x, y, and z axes. The x and y axes represent the horizontal plane, while the z axis represents the vertical direction. The values in 'position' and 'size' are in meters. The "caption" field contains textual description of the object generated by a vision-language model (VLM) based on several images of the object.

2. Bird's Eye View (BEV) Image: A top-down view of the scene, where objects' IDs are labeled in the image.

3. 2D Images: A set of 8 frames captured at equal intervals from the scene video. Each frame contains several objects with their object IDs labeled within red circles.

Note that object IDs are consistent across the scenegraph, 2D images, and BEV image.

Referring expressions are natural language descriptions that point to specific space within the 3D scene.

For example, a referring expression like "Facing the bed, move the nightstand 0.3 meters backward." requires identifying the 3D bounding box of the nightstand after it has been moved 0.3 meters backward.

Your task is to determine the position and size of the 3D bounding box corresponding to the referring expression.

After reviewing the information, I will provide a scene graph, 2D images, and a BEV image of a new scene, along with a referring expression. Your goal is to identify the 3D bounding box that corresponds to the referring expression.

At the end of your response, please provide the following details for the identified 3D bounding box:

1. The x, y, z coordinates of its center, strictly formatted as: xcoordinate: , ycoordinate: , zcoordinate:

2. The length of the 3D bounding box along the x, y, and z axes, strictly formatted as: {xlength: , ylength: , zlength: }

Figure B6: System Prompts of MLLMs textual inputs for space level

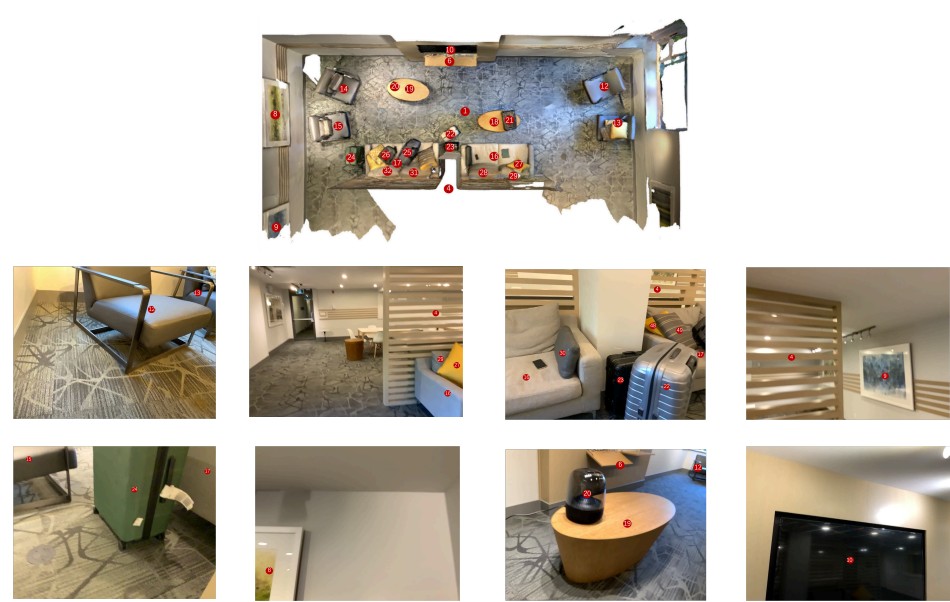

Figure B7: GPT4Scene Bird's Eye View (BEV) and eight uniformly sampled video frames from MultiScan Scene00109_00. The image at the top depicts the BEV, while the 2 × 4 grid below shows the video frames. In both the BEV and the video frames, object labels are marked with red circles to indicate object locations, following the visual input in GPT4Scene.

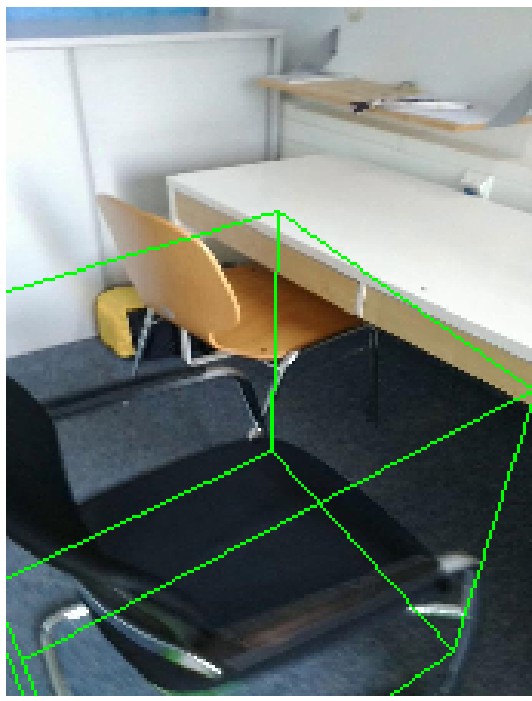

Figure B8: An example of Qwen-generated caption: "A black plastic chair with a cushioned seat and backrest. The frame is metallic, featuring curved legs and armrests. Designed for ergonomic seating in office or classroom settings. Positioned near a desk with papers and books, suggesting a workspace environment."

Analyze the object in the green bounding box across multiple viewpoints. Generate a caption with:

1. **Category**:
- Use specific name if confident (e.g., "mug")
- Use "generic-type object" for uncertain cases (e.g., "container-type object")
- Flag low-confidence predictions with "possibly"

2. **Attributes**:
- Color patterns & material properties
- 3D shape characteristics & structural features
- Functional affordances
- Contextual placement & surrounding objects

3. **Confidence Signals**:
- "The tapered rim suggests..."
- "While resembling a vase, the presence of..."
- "Inconclusive evidence for..."

Output template:
"A [material] [confidence][category] with [color] and [shape]. [Structure/texture details]. [Function inference]. [Contextual placement]."

Examples:
1. High Confidence: "A glossy ceramic mug with deep blue coloring and rounded shape. Has a comfortable handle and flat base. Made for holding hot drinks. Sitting beside a coffee maker and jar of beans on a kitchen counter."

2. Moderate Confidence: "A possibly glass vase-type object with translucent amber coloring. Fluted body shape and water droplets on surface. Likely floral display container. Found on windowsill with plants and pruning shears nearby."

3. Low Confidence: "A metallic tool-type object, matte silver with angular grooves. Ambiguous function between wrench or specialized clamp. Seen in workshop environment near assembly parts."

Figure B9: Prompts for Qwen generating object captions.

## B.4 Main Results

Here we report Acc@0.5IoU and Acc@0.75IoU on *Anywhere3D-Bench* respectively, as presented in Table B3 and Table B4.

Table B3: Results are presented in Acc@0.5IoU on Anywhere3D-Bench. *object bbox* in the table denotes the ground-truth object locations and sizes for simplicity. Chat-Scene*, Grounded 3D-LLM*: evaluations conducted on ScanNet. **Human performance is evaluated on a subset of 200 expressions obtained through stratified random sampling across four levels.

| | Open Source | Area Level | Space Level | Object Level | Part Level | Overall |
|---|---|---|---|---|---|---|
| **LLMs:** *object bbox, captions* | | | | | | |
| **non-thinking** | | | | | | |
| GPT-4.1 | ✗ | 51.85 ± 1.50 | 10.21 ± 0.17 | 45.81 ± 0.30 | 10.02 ± 1.19 | 24.67 ± 0.15 |
| Qwen2.5-72B | ✓ | 25.22 ± 1.22 | 3.75 ± 0.17 | 31.90 ± 0.85 | 1.75 ± 0.60 | 14.09 ± 0.47 |
| Qwen2.5-VL-72B | ✓ | 25.14 ± 1.12 | 3.02 ± 0.06 | 28.09 ± 1.48 | 1.68 ± 0.53 | 12.51 ± 0.69 |
| **thinking** | | | | | | |
| o4-mini | ✗ | 53.70 ± 2.62 | 11.66 ± 0.35 | 46.60 ± 0.22 | 10.39 ± 0.13 | 25.73 ± 0.23 |
| Qwen3-32B | ✓ | 30.51 ± 1.62 | 7.47 ± 0.26 | 38.40 ± 0.42 | 6.24 ± 0.11 | 18.98 ± 0.20 |
| DeepSeek-R1-671B | ✓ | 45.50 ± 1.40 | 9.15 ± 0.24 | 45.98 ± 0.24 | 7.68 ± 0.18 | 23.43 ± 0.20 |
| **MLLMs:** *object bbox, captions, BEV, video frames* | | | | | | |
| **non-thinking** | | | | | | |
| GPT-4.1 | ✗ | 54.50 ± 0.74 | 12.04 ± 0.06 | 51.36 ± 0.89 | 13.29 ± 0.53 | 28.05 ± 0.27 |
| LLaVA-OneVision-7B | ✓ | 4.76 | 1.08 | 7.02 | 0.38 | 3.16 |
| Qwen2.5-VL-72B | ✓ | 24.52 ± 1.10 | 5.60 ± 0.26 | 38.89 ± 0.36 | 3.43 ± 0.22 | 18.74 ± 0.22 |
| GPT4Scene | ✓ | 4.23 | 4.14 | 24.21 | 2.81 | 10.53 |
| **thinking** | | | | | | |
| o4-mini | ✗ | 54.50 ± 1.50 | 14.47 ± 0.12 | 53.78 ± 1.34 | 17.04 ± 0.27 | 30.54 ± 0.32 |
| **3D visual grounding models:** *point clouds, video frames* | | | | | | |
| PQ3D | ✓ | 11.82 ± 1.33 | 4.36 ± 0.13 | 23.52 ± 0.43 | 2.00 ± 0.39 | 10.74 ± 0.04 |
| 3D-VisTA | ✓ | 12.35 ± 0.31 | 3.89 ± 0.16 | 23.86 ± 0.48 | 2.06 ± 0.65 | 10.71 ± 0.16 |
| Chat-Scene* | ✓ | 27.24 ± 0.62 | 3.28 ± 0.31 | 30.12 ± 0.48 | 2.85 ± 0.00 | 16.39 ± 0.21 |
| Grounded 3D-LLM* | ✓ | 25.37 | 3.44 | 23.53 | 2.50 | 13.62 |
| Human** | – | 100.00 | 82.00 | 98.00 | 97.00 | 91.00 |

## B.5 Thinking v.s. Non-thinking

We also test Deepseek-V3 as well as Qwen3 non-thinking modes for comparison between thinking models and non-thinking models. As shown in Table B5, Qwen3-32B thinking modes outperform non-thinking modes consistently, and DeepSeek-R1-671B outperforms DeepSeek-V3-671B consistently. These results underscore the importance of reasoning capabilities in effectively addressing our benchmark.

## B.6 Ablation Study on Object Captions

In our evaluations of LLMs and MLLMs, textual captions for each objects are generated by Qwen2.5-VL-72B, one of the strongest **open-source** vision-language models available at the time. Our motivation for using Qwen2.5-VL-72B as the captioner lies in its open-source availability and to ensure reproducibility.

Nevertheless, the choice of captioner is flexible and can be replaced with other vision-language models, including proprietary ones. To explore the impact of different captioners on the final visual grounding performance, we additionally generate object captions using GPT-4.1 and compare the results to our original experimental setting, where Qwen2.5-VL-72B serves as the captioner. The following Table B6 shows the evaluation results in terms of Acc@0.25IoU on the human evaluation subset.

As shown, when GPT-4.1 is used as the captioner, it still remains among the top-performing models, while Qwen2.5-VL-72B still yields the lowest performance in the table. Also, utilizing models with stronger visual and spatial understanding abilities as captioner (GPT-4.1) result in better overall performance across all benchmarked models compared to Qwen2.5-VL-72B.

Table B4: Results are presented in Acc@0.75IoU on Anywhere3D-Bench. *object bbox* in the table denotes the ground-truth object locations and sizes for simplicity. Chat-Scene*, Grounded 3D-LLM*: evaluations conducted on ScanNet. **Human performance is evaluated on a subset of 200 expressions obtained through stratified random sampling across four levels.

| | Open Source | Area Level | Space Level | Object Level | Part Level | Overall |
|---|---|---|---|---|---|---|
| **LLMs: *object bbox, captions*** | | | | | | |
| **non-thinking** | | | | | | |
| GPT-4.1 | ✗ | 26.98 ± 0.74 | 4.05 ± 0.12 | 40.98 ± 0.29 | 4.68 ± 0.27 | 17.88 ± 0.25 |
| Qwen2.5-72B | ✓ | 5.11 ± 1.86 | 0.99 ± 0.22 | 28.76 ± 1.13 | 1.18 ± 0.11 | 10.47 ± 0.52 |
| Qwen2.5-VL-72B | ✓ | 6.08 ± 0.37 | 0.87 ± 0.06 | 25.47 ± 1.48 | 1.12 ± 0.00 | 9.39 ± 0.54 |
| **thinking** | | | | | | |
| o4-mini | ✗ | 30.69 ± 0.00 | 4.38 ± 0.12 | 42.19 ± 0.52 | 5.25 ± 0.27 | 18.76 ± 0.17 |
| Qwen3-32B | ✓ | 11.99 ± 1.10 | 2.62 ± 0.31 | 34.98 ± 0.26 | 2.91 ± 0.26 | 13.99 ± 0.15 |
| DeepSeek-R1-671B | ✓ | 23.99 ± 1.53 | 3.89 ± 0.30 | 41.40 ± 0.21 | 4.06 ± 0.11 | 17.64 ± 0.13 |
| **MLLMs: *object bbox, captions, BEV, video frames*** | | | | | | |
| **non-thinking** | | | | | | |
| GPT-4.1 | ✗ | 27.51 ± 4.50 | 4.96 ± 0.24 | 45.91 ± 0.30 | 6.08 ± 0.40 | 20.18 ± 0.42 |
| LLaVA-OneVision-7B | ✓ | 0 | 0.25 | 5.56 | 0 | 1.94 |
| Qwen2.5-VL-72B | ✓ | 3.88 ± 1.10 | 1.88 ± 0.13 | 34.63 ± 0.24 | 2.00 ± 0.11 | 12.86 ± 0.10 |
| GPT4Scene | ✓ | 1.59 | 0.41 | 21.28 | 0.19 | 7.35 |
| **thinking** | | | | | | |
| o4-mini | ✗ | 32.54 ± 0.37 | 5.54 ± 0.35 | 48.42 ± 1.04 | 8.05 ± 0.00 | 21.93 ± 0.14 |
| **3D visual grounding models: *point clouds, video frames*** | | | | | | |
| PQ3D | ✓ | 4.23 ± 1.06 | 0.47 ± 0.13 | 20.20 ± 0.54 | 0.00 ± 0.00 | 7.15 ± 0.16 |
| 3D-VisTA | ✓ | 3.35 ± 0.31 | 0.33 ± 0.00 | 21.35 ± 0.40 | 0.00 ± 0.00 | 7.41 ± 0.12 |
| Chat-Scene* | ✓ | 12.19 ± 0.62 | 0.60 ± 0.00 | 26.54 ± 0.33 | 0.00 ± 0.00 | 12.51 ± 0.11 |
| Grounded 3D-LLM* | ✓ | 5.97 | 0.86 | 17.86 | 0 | 8.59 |
| Human** | – | 100.00 | 48.00 | 97.00 | 83.00 | 74.00 |

Table B5: Evaluation between thinking and non-thinking models on the *Anywhere3D-Bench*. Specifically, we compare Qwen3-32B in its thinking and non-thinking modes, as well as DeepSeek-R1 v.s. DeepSeek-V3. Results are reported in terms of Acc@0.25IoU, Acc@0.5IoU, and Acc@0.75IoU.

| | Open Source | Area Level | Space Level | Object Level | Part Level | Overall |
|---|---|---|---|---|---|---|
| **Acc@0.25IoU** | | | | | | |
| Qwen3-32B(non-thinking) | ✓ | 54.67 | 9.60 | 31.97 | 12.24 | 20.43 |
| Qwen3-32B(thinking) | ✓ | 59.79 | 12.57 | 40.18 | 16.48 | 25.51 |
| DeepSeek-V3-671B | ✓ | 61.38 | 9.81 | 41.06 | 15.61 | 24.59 |
| DeepSeek-R1-671B | ✓ | 71.96 | 14.61 | 47.76 | 20.92 | 30.49 |
| **Acc@0.5IoU** | | | | | | |
| Qwen3-32B(non-thinking) | ✓ | 20.46 | 5.46 | 30.40 | 3.69 | 14.36 |
| Qwen3-32B(thinking) | ✓ | 30.51 | 7.47 | 38.40 | 6.24 | 18.98 |
| DeepSeek-V3-671B | ✓ | 26.63 | 6.18 | 39.52 | 4.81 | 18.29 |
| DeepSeek-R1-671B | ✓ | 45.50 | 9.15 | 45.98 | 7.68 | 23.43 |
| **Acc@0.75IoU** | | | | | | |
| Qwen3-32B(non-thinking) | ✓ | 5.82 | 1.87 | 27.46 | 1.75 | 10.57 |
| Qwen3-32B(thinking) | ✓ | 11.99 | 2.62 | 34.98 | 2.91 | 13.99 |
| DeepSeek-V3-671B | ✓ | 8.82 | 2.23 | 35.50 | 2.18 | 13.65 |
| DeepSeek-R1-671B | ✓ | 23.99 | 3.89 | 41.40 | 4.06 | 17.64 |

Table B6: Performance of LLMs and MLLMs under captions generated by GPT-4.1 and Qwen2.5-VL-72B on human evaluation subset. For simplicity, Qwen in the table denotes Qwen2.5-VL-72B. Each model is evaluated independently across three runs, with mean values reported.

| | Overall | | Area | | Space | | Object | | Part | |
|---|---|---|---|---|---|---|---|---|---|---|
| | GPT-4.1 captions | Qwen captions | GPT-4.1 captions | Qwen captions | GPT-4.1 captions | Qwen captions | GPT-4.1 captions | Qwen captions | GPT-4.1 captions | Qwen captions |
| **LLMs:** *object bbox, captions* | | | | | | | | | | |
| GPT-4.1 | 38.17 | 32.00 | 80.00 | 71.11 | 20.39 | 14.90 | 48.09 | 43.81 | 44.44 | 33.33 |
| o4-mini | 41.00 | 33.17 | 64.45 | 60.00 | 27.06 | 14.51 | 50.48 | 46.67 | 46.67 | 41.11 |
| Qwen-32B | 30.00 | 25.50 | 51.11 | 33.33 | 14.90 | 12.94 | 42.86 | 41.43 | 32.22 | 20.00 |
| Qwen2.5-72B | 21.67 | 18.83 | 22.22 | 17.78 | 8.63 | 6.67 | 39.05 | 34.76 | 17.78 | 16.67 |
| Qwen2.5-VL-72B | 18.33 | 16.17 | 33.33 | 33.33 | 7.45 | 5.10 | 24.76 | 27.62 | 26.67 | 12.22 |
| DeepSeek-R1-671B | 37.33 | 27.12 | 75.55 | 50.00 | 21.17 | 10.30 | 50.48 | 41.07 | 33.33 | 30.83 |
| **MLLMs:** *object bbox, captions, BEV, video frames* | | | | | | | | | | |
| GPT-4.1 | 40.50 | 37.00 | 84.45 | 86.67 | 21.18 | 15.29 | 52.86 | 54.29 | 44.45 | 33.33 |
| o4-mini | 44.33 | 41.00 | 82.22 | 80.00 | 28.24 | 20.00 | 56.19 | 57.14 | 43.33 | 43.33 |
| Qwen2.5-VL-72B | 28.00 | 26.83 | 68.89 | 68.89 | 9.02 | 8.63 | 41.91 | 39.05 | 28.89 | 28.89 |

## B.7 Detailed Analysis on Area Level

At the area level, we further categorize referring expressions into two types. The first type is *Objects Combination*, where most of the relevant objects are explicitly mentioned in the expression. For example, *"Identify the conference area with the long rectangular wooden table surrounded by chairs, used for meetings and discussions."* This expression explicitly refers to the area formed by the table and surrounding chairs. The second type is *Commonsense Reasoning*, where models are required to apply commonsense knowledge to infer implicitly indicated objects before identifying the corresponding area. For instance, *"Choose the conference area suitable for holding face-to-face meetings."* the expression does not directly mention the objects, and the model must first deduce the components that constitute such a conference area.

Table B7: Analysis on area level

| | LLM setting | | | VLM setting | | |
|---|---|---|---|---|---|---|
| Model | Qwen2.5-72B | Qwen3-32B | DeepSeek-R1-671B | GPT-4.1 | o4-mini | Qwen2.5-VL-72B |
| Object Combination | 66.15 | 64.62 | 84.62 | **95.38** | 87.69 | 64.62 |
| Commonsense Reasoning | 58.06 | 53.23 | 66.94 | **76.61** | 72.58 | 53.22 |

Table B7 demonstrates the performance of six top models on two types of expressions at area level. All models exhibit weaker performance on expressions that necessitate commonsense reasoning for the initial identification of the target objects, illustrating their limitations in applying commonsense knowledge within 3D scenes.

## B.8 Evaluation Cost

The inference cost varies significantly across models, with DeepSeek-R1-671B costing approximately $40, GPT-4.1 $53, o4-mini $120, and Qwen3-32B around $9 per full evaluation on *Anywhere3D-Bench*.

## C How Can We Improve MLLMs' Ability on Multi-level Visual Grounding?

### C.1 Error Analysis of Bounding Box Predictions

For the incorrect bounding boxes predicted by GPT-4.1, we first aim to analyze whether the error primarily arises from inaccuracies in the size or the position of the bounding boxes, with particular focus on the two challenging visual grounding levels, i.e., space level and part level. We set both the predicted and ground-truth bounding box positions to the origin (0, 0, 0) and compute Acc@0.25IoU. This allows us to assess how many predictions fail primarily due to incorrect size estimation, as illustrated in Table C1

Table C1: Error analysis of GPT-4.1's bounding box predictions. We report $\mathrm{Acc@0.25IoU}$ under two settings: (1) full IoU considering both location and size ("location + size"), the same setting reported in our main paper, and (2) IoU computed by aligning both predicted and ground-truth boxes at (0, 0, 0) to evaluate only size accuracy ("size").

| Model | area | space | object | part | overall |
|---|---|---|---|---|---|
| GPT-4.1(location + size) | 81.48 | 19.03 | 53.88 | 25.85 | 35.90 |
| GPT-4.1(size) | 94.18 | 88.67 | 78.36 | 64.98 | 81.24 |

The evaluation results reveal that GPT-4.1 achieves relatively high accuracy in predicting bounding box sizes—88.85% at the space level and 64.51% at the part level. The substantial increase in accuracy—from 18.95% to 88.85% at the space level, and from 29.02% to 64.5% at the part level—demonstrates that the majority of GPT-4.1's failures at these two levels can be attributed to inaccurate localization rather than size estimation.

Furthermore, we evaluate cases in which the predicted bounding boxes completely deviate from the ground truth—i.e., there is no spatial intersection between the two. To quantify whether any overlap exists between the predicted and ground-truth bounding boxes, we compute the metric $\mathrm{Acc@>0, IoU}$, as reported in Table C2. Notably, at the space level, approximately 70% of the predictions exhibit no overlap with the ground truth. This finding highlights that even the top-performing model struggles to accurately comprehend and reason about spatial configurations at the space level.

Table C2: GPT-4.1 performance measured by $\mathrm{Acc@>0\,IoU}$. Under this metric, a prediction is considered correct if there is any non-zero intersection between the predicted bounding box and the ground-truth bounding box.

| Model | area | space | object | part | overall |
|---|---|---|---|---|---|
| GPT-4.1 | 90.48 | 31.44 | 65.00 | 55.90 | 50.92 |

## C.2   Object Orientation Generated by Orient-Anything

In this section, we detail the process of generating object orientations using Orient Anything.

For each object in the scene graph, we first leverage meta-information from the video to select video frames capturing it from different viewpoints. Then we feed these frames into Orient Anything, which outputs predicted azimuth angle between the camera pose and the object, along with corresponding confidence scores. Then, for each video frame of the object, we combine the camera extrinsic parameters, the azimuth angle predicted by Orient Anything, and the scene coordinate system to compute the object's orientation in that frame. The orientation is classified into one of five categories: "positive x", "positive y", "negative x", "negative y", or "not sure". The "not sure" label is assigned when (i) the object's orientation is clearly not aligned with x-axis or y-axis, (ii) the object has no meaningful orientation (e.g., a circular footstool), or (iii) the confidence score predicted by Orient Anything is below 0.9.

We then apply majority voting across all video frames of the object to determine its final orientation, selecting from one of the five predefined categories, i.e. "positive x", "positive y", "negative x", "negative y", or "not sure".

As illustrated in Fig. C1, the ground-truth orientation of chair-1 is negative x. We first utilize Orient Anything to achieve its orientation in each frame. Then decide its final orientation through majority voting of these frames.

## C.3   More results on Visual Perception and Relational Reasoning Enhancement

We apply visual perception enhancement (human-selected key frames) and relational reasoning enhancement (BEV images with coordinate axes and object orientations predicted by Orient Anything) **jointly** to evaluate model performance under both enriched input enhancements. In addition, we manually annotate the object orientations in the scenes and use them as alternative inputs to compare

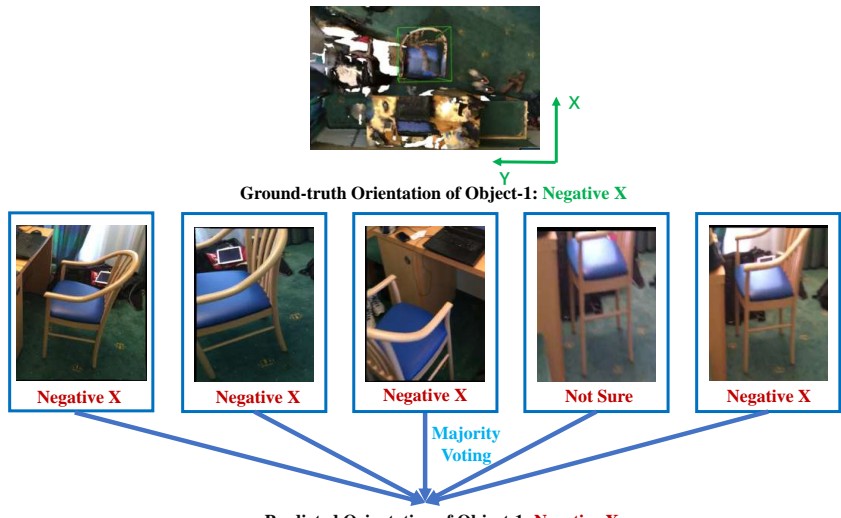

Figure C1: ground-truth orientation of object-1 and the predicted orientation of object-1 by Orient Anything.

model performance against predicted orientations from Orient-Anything, under the same enhancement settings.

Table C3: Effect of the visual perception enhancement and the relational reasoning enhancement, and visual perception and relational reasoning **jointly** . $\Delta$ denotes the change in accuracy relative to GPT-4.1. The reported results are based on the human evaluation subset and averaged across three independent trials.

| Method | Area | Space | Object | Part | Overall |
|---|---|---|---|---|---|
| *GPT-4.1* | 86.67 | 15.29 | 54.29 | 33.33 | 37.00 |
| $\Delta$(GPT-keyframe) | 4.44 ↓ | 2.75 ↑ | 2.86 ↓ | 8.90 ↑ | 1.00 ↑ |
| $\Delta$(Human-keyframe) | 6.66 ↑ | 7.85 ↑ | 3.33 ↑ | 15.57 ↑ | 7.50 ↑ |
| $\Delta$(BEV-axes) | 6.66 ↑ | 1.96 ↑ | 2.85 ↑ | 5.56 ↑ | 3.00 ↑ |
| $\Delta$(BEV-axes + ori.) | 2.22 ↑ | 5.49 ↑ | 0.96 ↓ | 3.34 ↑ | 4.00 ↑ |
| $\Delta$(Human-keyframe + BEV-axes + ori.) | 6.66 ↑ | 10.59 ↑ | 1.42 ↑ | 15.57 ↑ | 7.83 ↑ |
| $\Delta$(Human-keyframe + BEV-axes + human ori.) | 4.44 ↑ | 9.41 ↑ | 5.24 ↑ | 15.57 ↑ | 8.50 ↑ |

As shown in Table C3, applying both visual and relational enhancements leads to performance gains across all grounding levels compared to the baseline GPT-4.1. However, when compared to using only the *human-keyframe* visual enhancement, the addition of relational enhancement—even when using manually annotated orientations—does not yield a significant further improvement in performance at space level. Although object orientations and coordinate axes provide GPT-4.1 with richer spatial information, we suspect that the model struggles to correctly interpret and utilize this information. In particular, it appears to have difficulty reasoning about spatial relationships involving transformations between ego-centric views, object orientations, and the global coordinate system.

To further assess the model's understanding and reasoning of spatial information including object orientation, ego-centric views and global coordinate system, we design an experiment to evaluate GPT-4.1's ability to interpret object orientations and perform spatial relationship transformations. Specifically, we prompt the model with a question using the following template Fig. C2, where $axes \in \{+x, -x, +y, -y\}$ and $direction \in \{\text{left}, \text{right}\}$:

The ground-truth answers and those answers provided by GPT-4.1 are presented in Table C4. The results indicate that the model struggles with spatial relationship transformations. Even in cases where the final answer is correct, the underlying reasoning process is often flawed—for instance,

> Suppose the refrigerator's orientation is **axes** direction. If you are facing the refrigerator and using a right-handed coordinate system, which direction does your **direction** side point to (+x, -x, +y, or -y)?

Figure C2: Question template to test GPT-4.1 understanding on spatial information including orientation, ego-centric view and spatial relationship transformation.

Table C4: GPT-4.1's understanding on spatial orientations and spatial relationship transformations.

| orientation of the refrigerator | direction | ground-truth answer | GPT-4.1 answer | correct reasoning process? |
|---|---|---|---|---|
| -x | left | +y | +y | ✗ |
| -x | right | -y | -y | ✗ |
| +x | left | -y | +y | ✗ |
| +x | right | +y | -y | ✗ |
| -y | left | -x | -x | ✗ |
| -y | right | +x | -x | ✗ |
| +y | left | +x | -x | ✗ |
| +y | right | -x | +x | ✗ |

the model may arrive at the correct answer by making two offsetting transformation errors. This observation helps explain why GPT-4.1 continues to struggle with relational reasoning at the space level, even when provided with manually annotated object orientations and axes in BEV.

## C.4 Detailed Analysis on Qualitative Results

Here, we provide a detailed analysis of the qualitative results. As illustrated in Fig. C3, Fig. C4, Fig. C5, and Fig. C6, ground-truth bounding boxes are shown in green, while GPT-4.1's predicted bounding boxes are shown in red. In the "Reasoning" section below each example, the incorrect reasoning steps are underlined in red.

In Example (a), GPT-4.1 fails to correctly interpret the spatial relationship: the right side of the piano should correspond to the direction of decreasing x in the coordinate system. Additionally, the model places the small speaker on the floor rather than on top of the piano, which violates commonsense knowledge.

In Example (b), although GPT-4.1 roughly identifies the correct location of the clock, it misunderstands the orientation in which the clock is placed. The term *thickness* should refer to the extent to which the clock protrudes from the wall. A wall-mounted clock should be vertically aligned against the wall, rather than lying flat.

In Example (c), the primary mistake made by GPT-4.1 lies in its misinterpretation of the sofa's orientation. The orientation of a sofa should correspond to the direction a person would face when standing up from it. Additionally, the model made a minor error in identifying the starting point—it should be the midpoint of the front edge of the sofa, rather than its geometric center.

In Example (d), although GPT-4.1 correctly identifies the cabinet and recognized that the two compartments are distributed on the left and right sides along the y-axis, it still makes an error in spatial reasoning by failing to correctly map the concept of "left" and "right" to the corresponding directions of the coordinate axis.

In Example (e), GPT-4.1 simply treats the bottommost drawer as the cabinet(i.e., object-16). The lack of fine-grained visual details makes it difficult for the model to identify the white drawer located at the bottom of the cabinet.

In Example (f), GPT-4.1 makes errors in translating spatial relationships into coordinate-based representations and also fails to identify an object that is placed on the floor.

In Example(g), GPT-4.1 correctly identifies the two white chairs and its mathematical calculations are precise. However, the method it uses to compute the distance between the two objects in space is inappropriate. Rather than using the distance between their center points, the calculation should

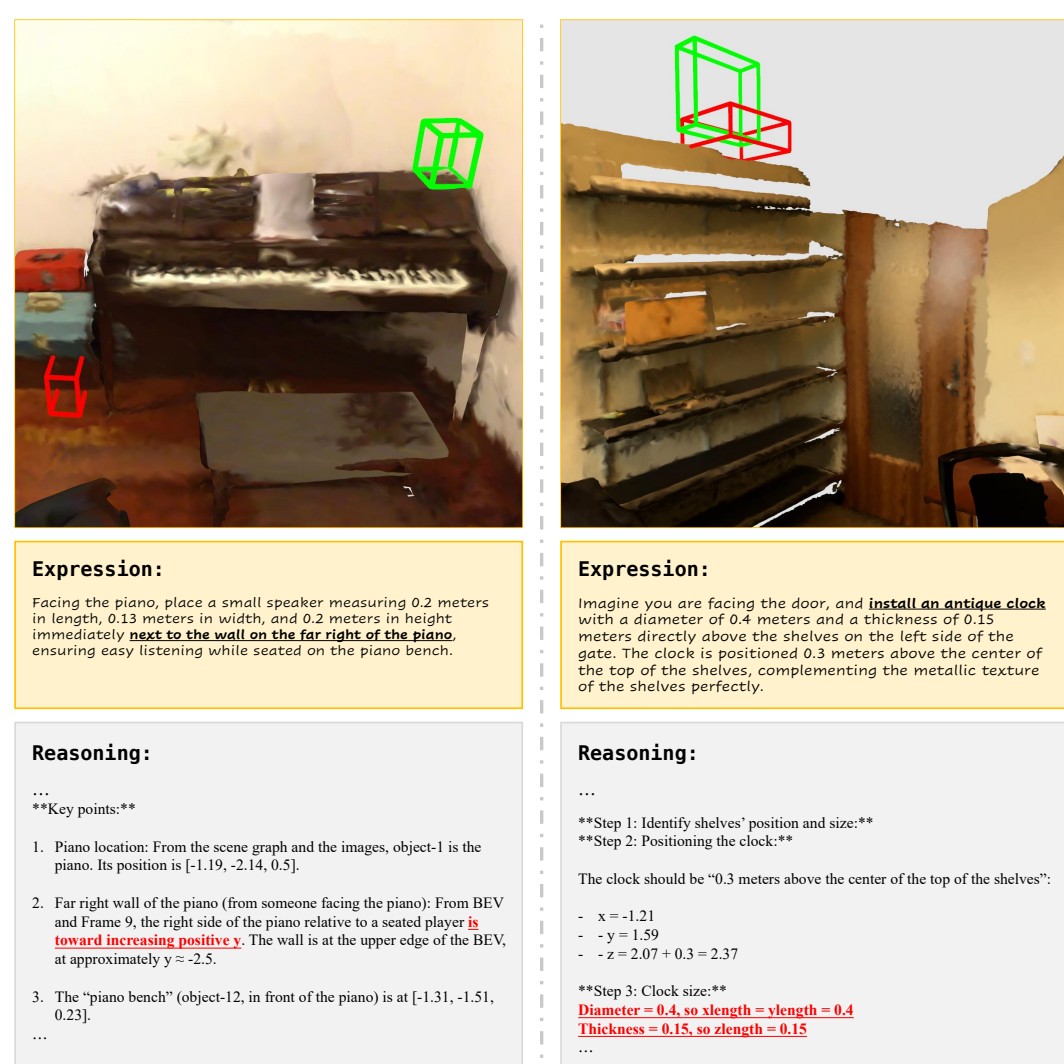

**Expression:**

Facing the piano, place a small speaker measuring 0.2 meters in length, 0.13 meters in width, and 0.2 meters in height immediately **next to the wall on the far right of the piano**, ensuring easy listening while seated on the piano bench.

**Expression:**

Imagine you are facing the door, and **install an antique clock** with a diameter of 0.4 meters and a thickness of 0.15 meters directly above the shelves on the left side of the gate. The clock is positioned 0.3 meters above the center of the top of the shelves, complementing the metallic texture of the shelves perfectly.

**Reasoning:**

...
**Key points:**

1. Piano location: From the scene graph and the images, object-1 is the piano. Its position is [-1.19, -2.14, 0.5].

2. Far right wall of the piano (from someone facing the piano): From BEV and Frame 9, the right side of the piano relative to a seated player **is toward increasing positive y**. The wall is at the upper edge of the BEV, at approximately y ≈ -2.5.

3. The "piano bench" (object-12, in front of the piano) is at [-1.31, -1.51, 0.23].

...

**Reasoning:**

...

**Step 1: Identify shelves' position and size:**
**Step 2: Positioning the clock:**

The clock should be "0.3 meters above the center of the top of the shelves":

-   x = -1.21
-   - y = 1.59
-   - z = 2.07 + 0.3 = 2.37

**Step 3: Clock size:**
**Diameter = 0.4, so xlength = ylength = 0.4**
**Thickness = 0.15, so zlength = 0.15**
...

Figure C3: Reasoning process of qualitative examples (a: left) (b: right).

follow commonsense reasoning—in this context, it should be the shortest distance between the two objects on the XY plane.,

In Example (h), GPT-4.1 correctly identifies sofa-1, coffee table-2, and sofa-4 along with their respective positions. However, a typical arrangement for 2–3 friends to sit, drink tea, and chat would usually involve being seated on the same sofa, since the distance between sofa-1 and sofa-4 exceeds 3.5 meters.

# D  Detailed Related Work

In this section, we provide a more detailed discussion of recent benchmarks related to spatial intelligence, including both task formatted as QA and visual grounding, and compare our proposed Anywhere3D-Bench with these benchmarks on four visual grounding levels.

**Object-level comparison with VSI-Bench:**

- VSI-Bench is formulated as a *question-answering* task, while Anywhere3D-Bench is a visual grounding benchmark.

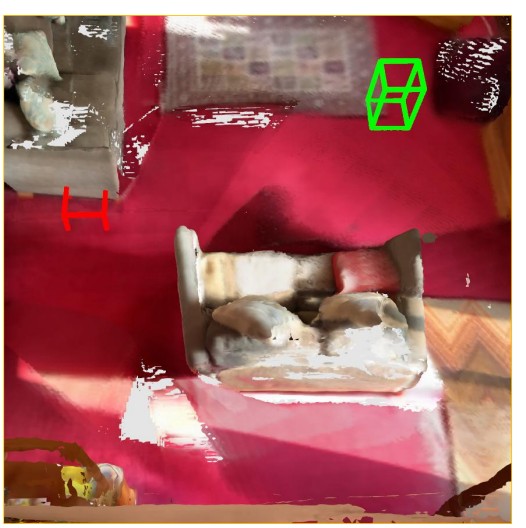

**Expression:**

Standing up from the middle of the sofa closer to the door, walk 1.5 meters **forward** and then **turn right**. Walk another 1 meter forward and stop. Place a trash bin with 0.3 meters in both width and length and 0.45 meters in height centered at the current position.

**Expression:**

Select the compartment of the cabinet below the window, where **the door opens to the right**.

**Reasoning:**

...

### 1. Identify the starting point:
**"the middle of the sofa closer to the door"**
-   The BEV shows the door is at the bottom-right, into the corridor/foyer (where object-10 is close by).
-   The sofa closer to this door is **object-1**, based on both position and BEV layout.

### 2. Find the center coordinate and orientation:
-   **Object-1 center:** (-0.11, 0.12, 0.64), size: [1.08, 1.63, 0.78]
-   The **middle** of the sofa is its center: (-0.11, 0.12, 0.64)

#### Orientation: What does "forward" mean?
-   "Walk forward" implies moving toward the front of the sofa if you are "standing up from it," most logically in the direction you'd be facing as you stand from sitting towards the room **(towards positive y).**
...

**Reasoning:**

...

1. **Identifying the Cabinet Below the Window:**
   - From the scene graph, object-8 is described as: "A white cabinet with a rectangular... Features two doors with vertical handles. Likely used for storage in the kitchen. Positioned below a countertop, adjacent to a sink and near other kitchen appliances."  matching object-8.

2. **Identifying the Correct Compartment (Right Door):**
   - Object-8, from both the BEV and 2D images, consists of two compartments (left and right doors). The rightmost compartment is the one whose door opens to the right.
   - Since object-8 has the position [2.0, 0.15, 0.58] and size [0.62, 0.98, 0.89], and has two doors, each door is roughly half the cabinet's y-size.

3. **Predicting the 3D Bounding Box for the Right Compartment:**
   - Center of the whole cabinet (object-8): [2.0, 0.15, 0.58].
   - Cabinet total y-length: 0.98 m. Each compartment y-length: 0.98 / 2 = 0.49 m.
   - **The right door is positioned at the higher y-value** (since in front view, right side = higher y, matching with other object placements in the BEV image).
...

Figure C4: Reasoning process of qualitative examples (c: left) (d: right).

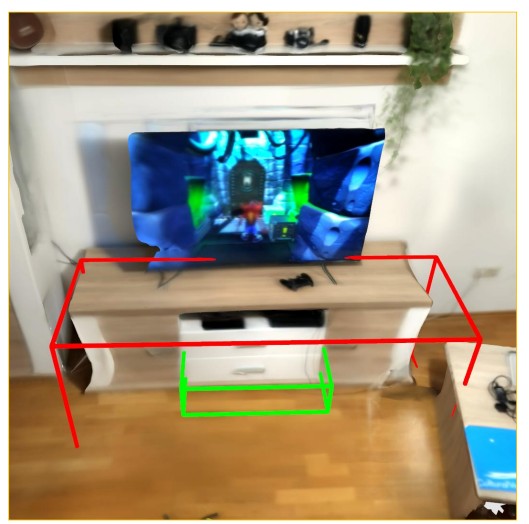

**Expression:**

Pull the white drawer at **the bottommost level** directly below the TV outward by 0.25 meters.

**Expression:**

Standing up from the armchair and walking 1 meter forward, which object on the floor **directly to my right** is against the wall?

**Reasoning:**

...
1. The TV is object-15, positioned at [0.12, -1.44, 2.66] with size [1.17, 0.19, 0.79].

2. Directly below the TV, the most likely candidate for a bottommost white drawer is object-16

3. The TV stand/cabinet, seen in the 2D images (frames 4 and 6), confirms the structure has drawers. **The bottommost drawer is the one closest to the floor, matching object-16.**

4. The referring expression says to pull the drawer outward by 0.25 meters. Outward here means in the +y direction (since drawers pull towards the viewer, who would be facing the TV from the main open part of the room/audience in the BEV).
...

**Reasoning:**

...
1. **Locate the armchair**:
   - From the BEV and scene graph, 'object-10' ("A woven wicker chair...") is the likely armchair. Its position is [-0.97, 1.85, 0.7].

2. **User walks 1 meter forward**:
   - Forward from the armchair (based on typical room orientation and BEV layout) would likely be decreasing y (since the opening toward the rest of the room is in that direction).
   - So, new approximate position: [-0.97, 0.85, 0.7]

3. **Object to the right, on the floor, and against the wall from this new position**:
   - "To the right" relative to the forward direction is toward increasing x.
   - Let's look at candidate objects near [-0.97, 0.85, 0.7] with greater x and next to the wall.
   Reviewing the scene graph, these are the possible candidates:
   - object-5: position [-1.8, 0.96, 1.05]
   - object-6: position [-1.81, 0.96, 1.84]
   - Now, with BEV confirmation:
   - The wall on the right after walking forward from the armchair (by 'object-10') would be the right-hand wall, where 'object-5' is closely positioned along the wall and **is on the floor (bookshelf base).**
...

Figure C5: Reasoning process of qualitative examples (e: left) (f: right).

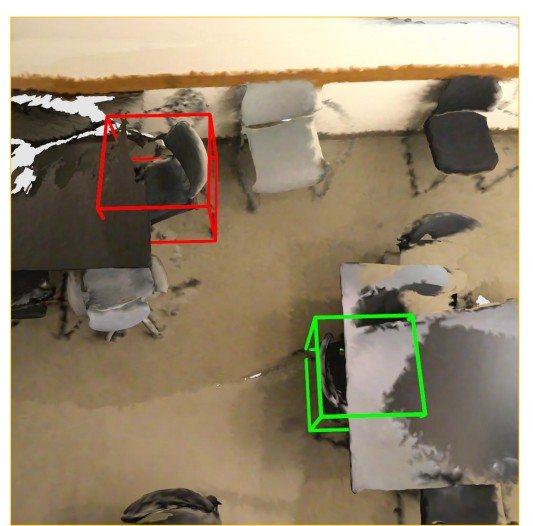

**Expression:**

Locate the black chair that **is 1 meter away** from **both** white chairs.

**Expression:**

Select an area suitable for **brewing tea, drinking tea**, and **chatting with two or three friends**.

**Reasoning:**

...
1. Identify the white chairs in the scene
Based on the scene graph, only two objects plausibly correspond to white chairs:
- **object-9**: ...
- **object-39**: ...

Positions:
- object-9: [-1.87, -0.45, 0.4]
- object-39: [-0.87, -1.48, 0.37]

2. Find the black chair that is 1 meter from both object-9 and object-39
Let's compute Euclidean distances (planar, ignoring z for simplicity, as z is the vertical):

For each black chair:
### Candidate 1: object-40 ([caption: dark gray upholstery])
- Position: [-1.53, -1.24]
**Distance** to object-9:
- $\sqrt{[(-1.53 - -1.87)^2 + (-1.24 - -0.45)^2]} = \sqrt{[(0.34)^2 + (-0.79)^2]} \approx \sqrt{(0.12 + 0.624)} \approx \sqrt{0.744} \approx 0.86$ m
**Distance** to object-39:
- $\sqrt{[(-1.53 - -0.87)^2 + (-1.24 - -1.48)^2]} = \sqrt{[(-0.66)^2 + (0.24)^2]} = \sqrt{(0.4356 + 0.0576)} \approx \sqrt{0.493} \approx 0.70$ m
So, object-40 is close to both, but not exactly 1 m.

...
### Candidate 10: object-24 ([caption: black coloring])
- Position: [-0.32, -0.08]
**Distance** to 9: [-0.32 + 1.87 = 1.55], [-0.08 + 0.45 = 0.37]
- $\sqrt{[2.4025 + 0.1369]} \approx \sqrt{2.539} \approx 1.59$ m
**Distance** to 39: [-0.32 + 0.87 = 0.55], [-0.08 + 1.48 = 1.4]
- $\sqrt{[0.3025 + 1.96]} \approx \sqrt{2.2625} \approx 1.50$ m
Still too far.

Given the ambiguity of scene size and natural tolerance, object-40 is best matching.
...

**Reasoning:**

...
**Requirements Breakdown:**

Space for making and drinking tea, and sitting comfortably together.
- Needs seating for 2 or 3 people, a coffee table for serving tea, and a relaxed social setting.

**Analysis:**
- The living room space in the BEV shows two sofas (object-1 and object-4), a coffee table (object-2), and a rug creating a gathering area.
- Object-2 (coffee table) in the scene graph is described as having a mug, a bowl of fruit, and a plate2014which fits the 201cbrewing and drinking teapurpose.
- **Object-1 and object-4 are beige sofas placed across and adjacent to the coffee table**, offering social seating.
- The surrounding area (including the rug) forms the ideal interaction space for relaxing and chatting, rather than the kitchen area at the top left with the fridge and appliances.
...

Figure C6: Reasoning process of qualitative examples (g: left) (h: right).

- Besides, questions in VSI-Bench are constructed using eight **strictly formatted question templates** (see Table 4 in VSI-Bench's appendix). For example, object size questions in VSI-Bench are limited to a fixed form asking about "the length of the longest dimension", whereas Anywhere3D features more diverse queries about objects, such as objects' aspect ratio and occupied floor area (as exemplified in Fig. 2).

**Part-level comparison with SceneFun3D:**

- Anywhere3D emphasizes the visual grounding of **part movements** by predicting the parts' positions after movement to test models' ability of moving parts in the 3D space (e.g., "pull the top drawer out until it touches the armchair" in Fig. 1), whereas SceneFun3D is designed to predict the *part's original positions and motion directions*.
- In addition, SceneFun3D focuses on 9 affordance categories of functional interactive elements of objects, such as "handles" and "knobs". In contrast, Anywhere3D involves more **open-ended object parts** (e.g., "toilet tank", "lampshade of the lamp", "top drawer of the cabinet").

**Space- and region-level comparison with ScanReason/Space3D-Bench/MMScan:**

Quite different from the spatial questions in other benchmarks, the "space-level" queries in Anywhere3D are intended to ground **unoccupied space** (as explained in Fig. 1's caption and Abstract), often involving placing a new object or moving an existing object to a specified unoccupied space within the scene, such as:

- *"Place a cup on the upper right corner of the bedside table."*(Fig. 1)
- *"Mount a clock on the wall above the shelf for convenient viewing."*(Fig. 6)
- *"Move the chair 0.5 meters backward."*

Comparatively, ScanReason focuses on object-level visual grounding; Space3D-Bench focuses on question-answering tasks concerning objects and rooms, whereas the area-level queries in Anywhere3D-Bench are not limited to rooms, but also include **functional areas**, possibly a portion of a room, e.g., the study area in Fig. 1. MMScan's region-level tasks are similar to our area-level tasks.

To clearly illustrate how Anywhere3D-Bench differs from prior benchmarks, we provide a detailed comparison in the Table D1 below:

Table D1: Comparison with recent benchmarks focusing on spatial intelligence

| Benchmark | Task Format | Area/Region/Room | Unoccupied Space | Object | Part |
|---|---|---|---|---|---|
| VSI-Bench | template-based QA | ✓ | ✗ | ✓ | ✗ |
| SceneFun3D | grounding | ✗ | ✗ | ✗ | ✓(only 9 functional interactive classes) |
| MMScan | grounding + QA | ✓ | ✗ | ✓ | ✗ |
| ScanReason | grounding | ✗ | ✗ | ✓ | ✗ |
| Space3D-Bench | grounding | ✓ | ✗ | ✓ | ✗ |
| **Anywhere3D-Bench(ours)** | grounding | ✓ | ✓ | ✓ | ✓ |

As shown, Anywhere3D-Bench provides a holistic evaluation for **multi-level** grounding in 3D scenes, while other benchmarks touch only one or two levels. Also, the **space-level** tasks, requiring reasoning about unoccupied space beyond objects, represent a particularly novel aspect of our benchmark.

# E   Limitations and Future Directions

In this section, we outline the limitations of our current work and propose directions for future research.

First, we plan to develop a training set for *Anywhere3D-Bench*. This would enable the supervised fine-tuning or reinforcement learning-based fine-tuning strategies to enhance the multi-level visual grounding capabilities of both 3D visual grounding models and lightweight VLMs.

Second, we aim to perform a deeper analysis of the reasoning processes produced by models and design corresponding evaluation metrics to assess the correctness of intermediate reasoning steps. In some cases, even if the final bounding box prediction is correct, the intermediate reasoning may

involve compensatory errors—e.g., two incorrect steps canceling each other out. To address this, we are hoping to construct dual-form expressions, such as converting "Facing the sofa, place a table on the right side of the sofa" into "Imagine sitting on the sofa, place a table on the left side." A model's prediction should only be considered correct if it answers both dual expressions consistently.

Third, while current model outputs are represented as the center coordinates and size of a 3D bounding box, we are interested in exploring a multiple-choice formulation. In this setting, the model would select the correct bounding box from a set of candidates, including the ground-truth and several distractors sampled from the scene.

Fourth, the proposed visual perception and relational reasoning enhancements serve as initial attempts to improve performance on Anywhere3D-Bench, highlighting the substantial gap between current models and humans. We hope these efforts will inspire future explorations, such as adopting video sampling strategies or other advanced techniques.

Fifth, we hope to move beyond visual grounding alone and explore object generation at the grounded location, such as generating corresponding 2D images of objects placed in the predicted 3D position.

Sixth, many expressions in *Anywhere3D-Bench* can naturally serve as instructions. In future work, we plan to extend the benchmark toward embodied tasks, enabling agents (e.g., robots or simulated avatars) to execute the instructions in interactive 3D environments.

