# OpenReview forum: "From Objects to Anywhere: A Holistic Benchmark for Multi-level Visual Grounding in 3D Scenes"
_NeurIPS.cc/2025/Datasets_and_Benchmarks_Track — NeurIPS 2025 Datasets and Benchmarks Track poster_

### Official Review · Reviewer_dPCa · 2025-06-03

**Rating:** 5
**Confidence:** 3

**Summary:**

This paper presents Anywhere3D-Bench, a comprehensive benchmark for multi-level 3D visual grounding that evaluates models across four hierarchical levels—area, space, object, and part—using 2,632 referring expression-3D bounding box pairs. Experimental results reveal that space-level (requiring complex 3D spatial reasoning) and part-level (needing fine-grained semantic perception) tasks are the most challenging. While multi-modal large language models (MLLMs) outperform traditional language models (LLMs) and specialized 3D visual grounding models, particularly in object and part-level tasks, they still struggle with spatial relational reasoning in 3D environments. The study also shows that enhancing inputs with global coordinate systems, object orientations, and key video frames can improve model performance on difficult tasks, though a substantial gap remains compared to human-level understanding. Overall, the benchmark highlights critical challenges in 3D scene comprehension beyond single-object grounding and provides insights for advancing multi-level 3D visual language capabilities.

**Dataset Code Accessibility:**

Yes

**Ethical Considerations:**

No, there are no or only very minor ethics concerns

**Final Justification:**

I appreciate the authors' thorough rebuttal, which effectively addresses my concerns. I recommend that the authors give further consideration to expanding the Anywhere3D-Bench through the plans in the rebuttal. I am pleased to raise my score to 5.

**Limitations Weaknesses:**

The most significant limitation is the relatively small dataset size (2,632 expression-bounding box pairs), especially for space-level (1,103) and part-level (386) categories. The complexity of multi-level tasks inherently requires more sample support, which may hinder models' ability to learn complex spatial relationships and fine-grained object structures. For example, compared to prior benchmarks like ScanRefer (5,410 expressions), the smaller scale might limit model generalization, as evidenced by the low accuracy of even the best-performing models on these levels (22.94% for space-level and 33.68% for part-level, as shown in Table 2). Expanding the dataset with more diverse scenes and expressions, or leveraging data augmentation techniques, could help address this limitation and improve model performance on challenging grounding levels.

**Strengths Contributions:**

The core contribution of this work is the creation of Anywhere3D-Bench, the first benchmark covering four hierarchical levels of 3D visual grounding—area, space, object, and part—using 2,632 referring expression-3D bounding box pairs to systematically evaluate models' multi-level scene understanding. Experiments reveal significant bottlenecks in space-level tasks (requiring complex 3D spatial reasoning, while multimodal large language models (MLLMs) outperform traditional language models (LLMs) and specialized 3D grounding models, particularly in object/part-level tasks—though their 3D spatial relational reasoning remains suboptimal. Additionally, incorporating structured spatial cues (global coordinates, object orientations) and key video frames boosts performance, yet a notable gap from human performance persists. Distinct from prior benchmarks like ScanRefer, Anywhere3D-Bench uniquely addresses diverse scene language (functional areas, trajectory-based instructions), providing a critical evaluation framework for embodied AI in AR/VR and robotics. Its open dataset and rigorous methodology further enhance community accessibility and impact.

---

> ### Author Rebuttal · Authors · 2025-07-30
>
> We sincerely appreciate your valuable feedback. Below, we provide detailed responses to your comments and concerns, and we hope our clarifications will address the issues you raised.
>
> > **Q1:** The most significant limitation is the relatively small dataset size (2,632 expression-bounding box pairs), especially for space-level (1,103) and part-level (386) categories. The complexity of multi-level tasks inherently requires more sample support, which may hinder models' ability to learn complex spatial relationships and fine-grained object structures. For example, compared to prior benchmarks like ScanRefer (5,410 expressions), the smaller scale might limit model generalization, as evidenced by the low accuracy of even the best-performing models on these levels (22.94% for space-level and 33.68% for part-level, as shown in Table 2). Expanding the dataset with more diverse scenes and expressions, or leveraging data augmentation techniques, could help address this limitation and improve model performance on challenging grounding levels.
>
> Thank you for your constructive suggestion regarding the dataset size. We would like to address your concern from several perspectives.
>
> **Current scale and effort:**
>
> As listed in Table 1 of our paper, Anywhere3D-Bench's data scale (2,632 expressions) is larger than that of SceneFun3D[1] (1,265 expressions) and ScanReason[2] (1,474 expressions). Annotating Anywhere3D-Bench requires considerable human effort, with each expression undergoing an average modification ratio of 42% to ensure correctness.
>
> **Dataset expansion plans:**
>
> Nevertheless, we agree with the reviewer that expanding the dataset would be beneficial. We plan to further increase the quantity of data through several approaches, including:
>
>  1. utilizing more advanced MLLMs (e.g., OpenAI o3, Google Gemini-2.5-Pro) to generate higher-quality referring expressions.
>  2. leveraging finer-grained annotations introduced in recent works (e.g., [3], [4]) on existing scene datasets such as ScanNet++ and ARKitScenes to further increase data volume. These works provide more comprehensive and detailed annotations for objects and their language descriptions. For example, ExCap3D[3] introduces part-level detailed captions for objects in ScanNet++.
>  3. applying data augmentation techniques, such as paraphrasing, to enrich expression diversity. For example, referring expressions like "*Facing the sofa, place a table on the right side of the sofa*" can be paraphrased as "*Imagine sitting on the sofa, place a table on the left side.*"
>
> [1] *SceneFun3D: Fine-Grained Functionality and Affordance Understanding in 3D Scenes (CVPR 2024)*
>
> [2] *ScanReason: Empowering 3D Visual Grounding with Reasoning Capabilities (ECCV 2024)*
>
> [3] *ExCap3D: Expressive 3D Scene Understanding via Object Captioning with Varying Detail (ICCV 2025)*
>
> [4] *Cubify Anything: Scaling Indoor 3D Object Detection (CVPR 2025)*
>
>
>
>
> We hope that our responses have addressed your questions and concerns. Please feel free to let us know if you have any further questions or concerns!

---

> > ### Comment · Reviewer_dPCa · 2025-08-09
> >
> > I appreciate the authors' thorough rebuttal, which effectively addresses my concerns. I recommend that the authors give further consideration to expanding the Anywhere3D-Bench through the plans in the rebuttal. I am pleased to raise my score to 5.

---

> > > ### Author Response · Authors · 2025-08-09
> > >
> > > Dear Reviewer dPCa,
> > >
> > > &emsp;&emsp; We would like to sincerely thank you again for your valuable suggestion, as well as your recognition and encouragement regarding our work. If you have any further questions or suggestions, please feel free to reach out to us at any time.

---

### Official Review · Reviewer_r58L · 2025-06-14

**Rating:** 5
**Confidence:** 4

**Summary:**

This paper introduces Anywhere3D-Bench, a new benchmark for multi-level 3D visual grounding that contains four hierarchical levels: area, space, object, and part. The benchmark contains 2,632 referring expression-3D bounding box pairs across 276 scenes from multiple datasets. The authors evaluate various state-of-the-art models including LLMs, MLLMs, and specialized 3D visual grounding models, revealing significant challenges in space-level and part-level tasks where even the best-performing model (o4-mini) achieves only 22.94% and 33.68% accuracy respectively.

**Dataset Code Accessibility:**

Yes

**Dataset Code Comments:**

The related method in this paper can be easily reproduced according to this paper.

**Ethical Considerations:**

No, there are no or only very minor ethics concerns

**Final Justification:**

The authors' rebuttal resolved my concerns. So, I decide to raise my score.

**Limitations Weaknesses:**

(1) The data size of Anywhere3D-Bench is still small. If possible, it is recommended to collect more data to support 3D visual grounding.

(2) The reliance on GPT-4 for initial expression generation may introduce biases from the language model's training data. While human verification helps, the paper doesn't quantify how much the final expressions diverge from the original expression generated by GPT.

(3) The distribution across levels is uneven (Space: 1,103 vs Part: 386), which may affect performance of different tasks, particularly for the part-level task.

(4) In the metric, the fixed IoU threshold of 0.25 may not be equally appropriate for all four levels. Part-level task might require stricter thresholds.

**Strengths Contributions:**

(1) Anywhere3D-Bench contains four hierarchical levels: area, space, object, and part, which is meaningful for 3D visual grounding.

(2) The space-level tasks, which require reasoning about unoccupied space and spatial relationships beyond objects, represent a particularly novel and challenging aspect of the benchmark.

(3) The extensive evaluation of diverse models (e.g., LLMs, MLLMs, and specialized 3D models) provides valuable insights into current capabilities and limitations in 3Dl visual grounding.

---

> ### Author Rebuttal · Authors · 2025-07-30
>
> We sincerely appreciate your valuable feedback. Below, we provide detailed responses to your comments and concerns, and we hope our clarifications will address the issues you raised.
>
> > **Q1:** The data size of Anywhere3D-Bench is still small. If possible, it is recommended to collect more data to support 3D visual grounding.
>
> Thank you for your constructive suggestion regarding the dataset size. We would like to address your concern from several perspectives.
>
> **Current scale and effort:**
>
> As listed in Table 1 of our paper, Anywhere3D-Bench's data scale (2,632 expressions) is larger than that of SceneFun3D[1] (1,265 expressions) and ScanReason[2] (1,474 expressions). Annotating Anywhere3D-Bench requires considerable human effort, with each expression undergoing an average modification ratio of 42% to ensure correctness (please see our response to your question **Q2** for details).
>
> **Dataset expansion plans:**
>
> Nevertheless, we agree with the reviewer that expanding the dataset would be beneficial. We plan to further increase the quantity of data through several approaches, including:
>
>  1. utilizing more advanced MLLMs (e.g., OpenAI o3, Google Gemini-2.5-Pro) to generate higher-quality referring expressions.
>  2. leveraging finer-grained annotations introduced in recent works (e.g., [3], [4]) on existing scene datasets such as ScanNet++ and ARKitScenes to further increase data volume. These works provide more comprehensive and detailed annotations for objects and their language descriptions. For example, ExCap3D[3] introduces part-level detailed captions for objects in ScanNet++.
>  3. applying data augmentation techniques, such as paraphrasing, to enrich expression diversity. For example, referring expressions like "*Facing the sofa, place a table on the right side of the sofa*" can be paraphrased as "*Imagine sitting on the sofa, place a table on the left side.*"
>
> [1] *SceneFun3D: Fine-Grained Functionality and Affordance Understanding in 3D Scenes (CVPR 2024)*
>
> [2] *ScanReason: Empowering 3D Visual Grounding with Reasoning Capabilities (ECCV 2024)*
>
> [3] *ExCap3D: Expressive 3D Scene Understanding via Object Captioning with Varying Detail (ICCV 2025)*
>
> [4] *Cubify Anything: Scaling Indoor 3D Object Detection (CVPR 2025)*
>
>
> > **Q2:** The reliance on GPT-4 for initial expression generation may introduce biases from the language model's training data. While human verification helps, the paper doesn't quantify how much the final expressions diverge from the original expression generated by GPT.
>
> Thank you for your kind reminder. Here, we provide quantitative details on the divergence of final expressions from the original expression generated by GPT.
>
> Before annotation began, we established clear guidelines: human annotators are allowed to revise the GPT-4o-generated expressions, but they first had to estimate the proportion of modification within the expression. **If the required changes exceeded 50% of the original expression**, annotators were allowed to ''skip'' that referring expression. Expressions needing such extensive revision were filtered out.
>
> Overall, **25%** of the candidate expressions were marked as ''skip'' and thus excluded from the dataset. For the remaining expressions, the Average Modification Ratio is approximately **42%**, as calculated by:
>
>
> $$
> \text{Average Modification Ratio} = \frac{1}{N} \sum_{i=1}^{N} \frac{\text{Levenshtein}(E_i^{\mathrm{GPT}}, E_i^{\mathrm{Human}})}{|E_i^{\mathrm{GPT}}|}
> $$
>
> where $N$ denotes the total number of referring expressions, $\text{Levenshtein}(E_i^{\mathrm{GPT}}, E_i^{\mathrm{Human}})$ represents the Levenshtein distance (i.e., the minimum number of single-word edits required to transform the GPT-4o-generated expression $E_i^{\mathrm{GPT}}$ into the final human-annotated expression $E_i^{\mathrm{Human}}$), and $|E_i^{\mathrm{GPT}}|$ is the length of the original GPT-4o-generated expression.
>
>
>
>
>
>
> > **Q3:** The distribution across levels is uneven (Space: 1,103 vs Part: 386), which may affect performance of different tasks, particularly for the part-level task.
>
> As mentioned in the response to your question **Q1** above, we are exploring ways to further increase the quantity of Anywhere3D-Bench. Specifically, at the part level, we have noticed that some concurrent work (e.g., ExCap3D, ICCV 2025) has introduced finer-grained part annotations. We will consider integrating these annotations and adding more part-level expressions to our dataset.
>
>
> > **Q4:** In the metric, the fixed IoU threshold of 0.25 may not be equally appropriate for all four levels. Part-level task might require stricter thresholds.
>
> We have provided evaluation results for IoU thresholds 0.5 and 0.75 across all four grounding levels in our Supplementary Material (*Table B3* and *Table B4*) due to the limited space in the main paper. Here, we list out the top-performing models' performance on the part-level for clarification in the following table:
>
> |||Acc@0.25IoU|Acc@0.5IoU|Acc@0.75IoU|
> |--|--|----------|---------|----------|
> |**LLMs**|GPT-4.1|25.52|11.52|5.70|
> ||o4-mini|26.94|11.53|6.22|
> ||Qwen3-32B|19.17|7.08|3.25|
> ||Qwen2.5-72B|11.40|2.42|1.64|
> ||DeepSeek-R1-671B|24.79|9.58|5.09|
> |**MLLMs**|GPT-4.1|29.02|15.02|7.38|
> ||o4-mini|33.68|17.62|8.81|
> ||Qwen2.5-VL-72B|17.79|4.75|2.76|
> |**3D-VG-models**|PQ3D|20.90|2.76|0.00|
> ||3D-VisTA|20.64|2.85|0.00|
> ||Chat-Scene|25.31|3.74|0.00|
> |**Human**|-|97.00|97.00|83.00|
>
>
> The table reveals two key observations, which are consistent with the findings discussed in our paper:
>
> - **Part-level grounding is particularly challenging**, requiring fine-grained perception of object composition and precise localization, where current models still fall short. As the IoU threshold increases, the performance of all models drops significantly on the part-level tasks, while human performance remains high and stable.
> - Overall, **MLLMs outperform 3D visual grounding specialist models**, as they are better at leveraging visual information and commonsense knowledge to localize the parts of objects referred in the expressions.
>
> We will also consider adopting a stricter IoU threshold for part-level tasks in our revised manuscript.
>
>
> We hope that our responses have addressed your questions and concerns. Please feel free to let us know if you have any further questions or concerns!

---

> > ### Comment · Reviewer_r58L · 2025-08-03
> >
> > Thanks for authors' rebuttal, which resolves my concerns. I encourage authors to further increase the quantity of Anywhere3D-Bench. And I'm pleased to raise my score to 5.

---

> > > ### Author Response · Authors · 2025-08-03
> > >
> > > Dear reviewer r58L,
> > >
> > > &emsp;&emsp; We would like to sincerely thank you again for your valuable feedback and your recognition and encouragement regarding our work. If you have any further questions or suggestions, please feel free to reach out to us at any time.

---

### Official Review · Reviewer_kMLK · 2025-07-02

**Rating:** 4
**Confidence:** 5

**Summary:**

The paper presents a new benchmark for 3D visual grounding in indoor scenes, comprising 2,632 referring expressions that span four levels of granularity: object part, object, space, and area. They evaluate three types of pretrained models on this benchmark: a). LLMs with text input alone, b). MLLMs with both text and images/videos, and c). expert 3D visual-grounding models. Finally, they discuss two strategies for boosting MLLM performance on such tasks: better video-frame sampling and the inclusion of explicit object-orientation cues, showing both to provide measurable gains.

**Additional Feedback:**

The annotation interface looks very similar to ScanRefer’s. If it is modified from ScanRefer, this should be clearly acknowledged.

**Dataset Code Accessibility:**

Yes

**Dataset Code Comments:**

The dataset is publicly accessible on HuggingFace, and the code is released on GitHub. Technical details are also well discussed in the supplementary materials.

**Ethical Comments:**

The proposed benchmark and dataset are constructed using existing publicly available 3D indoor scene datasets, including ScanNet, MultiScan, 3RScan, and ARKitScenes. As such, no significant ethical concerns remain.

**Ethical Considerations:**

No, there are no or only very minor ethics concerns

**Final Justification:**

The rebuttal addresses some of my concerns. I encourage the authors to incorporate the extra explanations and the two tables above into the paper, as these would help clarify the paper’s position and ensure a fairer experimental setting. I now recognize that the work differs from prior approaches in several aspects and offers new perspectives for 3D scene understanding. Accordingly, I am raising my rating to 4.

**Limitations Weaknesses:**

1. The paper presents a four-level (part, object, space, and area) 3D grounding benchmark as its primary contribution. However, this appears to be a straightforward aggregation of existing grounding and reasoning tasks rather than a fundamentally novel formulation. Prior works have already addressed each level individually — for example, VSI-Bench covers object-level QA and MCQ tasks, SceneFun3D addresses part-level grounding, and MMScan / ScanReason / Space3D-Bench tackle space- and region-level grounding. The paper lacks a clear discussion of how its benchmark meaningfully advances or differs from these existing efforts.
2. The proposed visual perception enhancement strategy (Sec 4.1) aimed at improving MLLM performance on multi-level visual grounding has also been explored in prior work. For instance, Video-3D LLM and Lexicon3D analyze various video sampling strategies and address the problem of selecting informative object-centric frames (i.e., the "maximum coverage" issue). The current paper does not sufficiently distinguish its approach from these related methods, nor does it clarify its novel contributions in this regard.
3. The paper generates object captions using Qwen2.5-VL-72B to serve as textual input for LLMs and MLLMs. This raises a potential concern about fairness: Qwen2.5-VL-72B may be at a disadvantage since it is evaluated on outputs derived from its own capabilities, but it may benefit other models by providing additional object information that is not "new" to Qwen itself. This issue needs further clarification and justification in the experimental design.

**Strengths Contributions:**

1. The paper introduces multi-level 3D visual grounding tasks that assess models' spatial understanding and reasoning capabilities in greater depth.
2. The benchmark includes a carefully designed human verification and correction procedure, enhancing the reliability of both the dataset and evaluation results.
3. The paper provides details of the data construction pipeline and experimental design, contributing to the reproducibility of the work.

---

> ### Author Rebuttal · Authors · 2025-07-30
>
> We sincerely appreciate your valuable feedback. Below, we provide detailed responses to your comments and concerns, and hope our clarifications will address the issues you raised.
>
> > **Q1:** The paper presents a four-level (part, object, space, and area) 3D grounding benchmark as its primary contribution. However, this appears to be a straightforward aggregation of existing grounding and reasoning tasks rather than a fundamentally novel formulation. Prior works have already addressed each level individually — for example, VSI-Bench covers object-level QA and MCQ tasks, SceneFun3D addresses part-level grounding, and MMScan / ScanReason / Space3D-Bench tackle space- and region-level grounding. The paper lacks a clear discussion of how its benchmark meaningfully advances or differs from these existing efforts.
>
>
> We compare our proposed Anywhere3D-Bench with the aforementioned benchmarks on different levels and discuss the differences as below:
>
> **Object-level comparison with VSI-Bench**:
> - VSI-Bench is formulated as a *question-answering* task, while Anywhere3D-Bench is a *visual grounding* benchmark.
> -  Besides, questions in VSI-Bench are constructed using eight **strictly formatted question templates** (see Table 4 in VSI-Bench's appendix). For example, object size questions in VSI-Bench are limited to a fixed form asking about "the length of the longest dimension," whereas Anywhere3D features more diverse queries about objects, such as objects' aspect ratio and occupied floor area (as exemplified in Figure 2 of our paper).
>
> **Part-level comparison with SceneFun3D**:
>
> - Anywhere3D emphasizes the visual grounding of **part movements** by predicting the *parts' positions after movement* to test models' ability of moving parts in the 3D space (e.g., ''pull the top drawer out until it touches the armchair'' in Figure 1), whereas SceneFun3D is designed to predict the *part’s original positions and motion directions*.
> - In addition, SceneFun3D focuses on 9 affordance categories of functional interactive elements of objects, such as "handles" and "knobs". In contrast, Anywhere3D involves **more open-ended object parts** (e.g., ''toilet tank'', ''lampshade of the lamp'', ''top drawer of the cabinet'').
>
>
> **Space- and region-level comparison with ScanReason/Space3D-Bench/MMScan**:
>
> Quite different from the spatial questions in other benchmarks, the "space-level" queries in Anywhere3D are intended to ground **unoccupied space** (as explained in Figure 1's caption and Abstract), often involving placing a new object or moving an existing object to a specified unoccupied space within the scene, such as:
>  - *"Place a cup on the upper right corner of the bedside table."* (Figure 1)
>  - *"Mount a clock on the wall above the shelf for convenient viewing."* (Figure 6)
>  - *"Move the chair 0.5 meters backward."*
>
>
> Comparatively, ScanReason focuses on object-level visual grounding; Space3D-Bench focuses on question-answering tasks concerning objects and rooms, whereas the area-level queries in Anywhere3D-Bench are not limited to rooms, but also include **functional areas**, possibly a portion of a room, e.g., the study area in Figure 1. MMScan's region-level tasks are similar to ours.
>
>
> To clearly illustrate how Anywhere3D-Bench differs from prior benchmarks, we provide a detailed comparison in the table below:
>
> |Benchmark|Task Format|Area/Region/Room| Unoccupied Space|Object|Part|
> |-|-|-|-|-|-|
> |VSI-Bench|template-based QA|✔️ |❌|✔️| ❌|
> |SceneFun3D|Grounding|❌|❌|❌|✔️ (only 9 functional interactive classes)|
> |MMScan|Grounding + QA|✔️|❌|✔️| ❌|
> |ScanReason|Grounding|❌|❌|✔️| ❌|
> |Space3D-Bench|QA|✔️|❌|✔️| ❌|
> |**Anywhere3D-Bench (ours)**|Grounding|✔️|✔️|✔️|✔️|
>
> Considering these differences, we believe the proposed Anywhere3D-Bench is NOT a straightforward aggregation of existing benchmarks, but provides a holistic evaluation for multi-level grounding in 3D scenes, while other benchmarks touch only one or two levels. We will incorporate the above discussion and position our paper more appropriately in the revision.
>
>
>
> > **Q2:** The proposed visual perception enhancement strategy (Sec 4.1) aimed at improving MLLM performance on multi-level visual grounding has also been explored in prior work. For instance, Video-3D LLM and Lexicon3D analyze various video sampling strategies and address the problem of selecting informative object-centric frames (i.e., the "maximum coverage" issue). The current paper does not sufficiently distinguish its approach from these related methods, nor does it clarify its novel contributions in this regard.
>
>
> Different from the "maximum coverage" strategy used in Video-3D LLM and Lexicon3D, the proposed visual perception enhancement strategy is designed to provide the models with **more query-relevant visual cues**. As explained in Lines 237-242, we first prompt GPT-4.1 to propose relevant objects based on the query and then append video frames containing the proposed objects to the original inputs.
>
> We consider this visual perception enhancement, together with another relational reasoning enhancement, as initial attempts to improve the performance on Anywhere3D. Instead of a significant technical contribution, we view these attempts as an indicator of the large performance gap between current models and human, and we hope they would inspire more future explorations in this regard.
>
>
> > **Q3:** The paper generates object captions using Qwen2.5-VL-72B to serve as textual input for LLMs and MLLMs. This raises a potential concern about fairness: Qwen2.5-VL-72B may be at a disadvantage since it is evaluated on outputs derived from its own capabilities, but it may benefit other models by providing additional object information that is not "new" to Qwen itself. This issue needs further clarification and justification in the experimental design.
>
>
> To verify this issue, we additionally generate object captions using GPT-4.1 and compare these results to our original experimental setting, where Qwen2.5-VL-72B serves as the captioner. The following table shows the evaluation results in Acc@0.25IoU on the human evaluation subset.
>
>
> |||Overall||Area||Space||Object||Part||
> |-|-|-|-|-|-|-|-|-|-|-|-|
> |||GPT-4.1 cap.|Qwen cap.|GPT-4.1 cap.|Qwen cap.|GPT-4.1 cap.|Qwen cap.|GPT-4.1 cap.|Qwen cap.|GPT-4.1 cap.|Qwen cap.|
> |**LLM**|GPT-4.1|38.17|32.00| 80.00|71.11|20.39|14.90|48.09|43.81|44.44|33.33|
> ||o4-mini|41.00|33.17| 64.45|60.00|27.06|14.51|50.48|46.67|46.67|41.11|
> ||Qwen-32B|30.00|25.50| 51.11|33.33|14.90|12.94|42.86|41.43|32.22|20.00|
> ||Qwen2.5-72B|21.67|18.83| 22.22|17.78|8.63|6.67|39.05|34.76|17.78|16.67|
> ||Qwen2.5-VL-72B|18.33|16.17| 33.33|33.33|7.45|5.10|24.76|27.62|26.67|12.22|
> ||DeepSeek-R1-671B|37.33|27.12| 75.55|50.00|21.17|10.30|50.48|41.07|33.33|30.83|
> |**MLLM**|GPT-4.1|40.50|37.00| 84.45|86.67|21.18|15.29|52.86|54.29|44.45|33.33|
> ||o4-mini|44.33|41.00|82.22|80.00|28.24|20.00|56.19|57.14|43.33|43.33|
> ||Qwen2.5-VL-72B|28.00|26.83| 68.89|68.89|9.02|8.63|41.91|39.05|28.89|28.89|
>
> **When GPT-4.1 is used as the captioner, it remains among the top-performing models, while Qwen2.5-VL-72B still records the lowest performance in the table.** This suggests that the inferior results of Qwen2.5-VL-72B on Anywhere3D-Bench are mainly due to its limited visual and spatial understanding abilities, rather than a disadvantage from serving as the captioner to provide additional information to other models.
>
> Our motivation behind the experimental design and evaluation is to emphasize **open-source and reproducibility**. Accordingly, for caption generation, we utilized Qwen2.5-VL-72B, which was among the best open-source vision-language models available at the time. We will also consider incorporating the above results into our revised manuscript for better clarity.
>
> > **Q4:** The annotation interface looks very similar to ScanRefer’s. If it is modified from ScanRefer, this should be clearly acknowledged.
>
> Thank you for your kind reminder. Our annotation interface is indeed adapted from ScanRefer and we will clearly acknowledge it in our revised manuscript.
>
> We hope that our responses have addressed your questions and concerns. Please feel free to let us know if you have any further questions or concerns!

---

> > ### Author Response · Authors · 2025-08-04
> >
> > Dear Reviewer kMLK,
> >
> > &emsp;&emsp; As the discussion period is nearly halfway through, we would like to ask if the clarifications and the additional results in the rebuttal address your concerns. If more explanations or experiments are needed, please feel free to let us know.
> >
> > &emsp;&emsp; Thank you again for your valuable feedback!

---

> > > ### Comment · Reviewer_kMLK · 2025-08-04
> > >
> > > Thanks to the authors for the rebuttal, which addresses some of my concerns. I have no further questions, and I encourage the authors to incorporate the two additional tables from the rebuttal into the paper. I have raised my rating to 4.

---

> > > > ### Author Response · Authors · 2025-08-04
> > > >
> > > > Dear reviewer kMLK,
> > > >
> > > > &emsp;&emsp; We would like to sincerely thank you again for your valuable feedback, as well as for your recognition and encouragement of our work. We will further improve the manuscript for better clarification by incorporating the additional tables. If you have any further questions or suggestions, please feel free to reach out to us at any time.

---

### Official Review · Reviewer_ZXVe · 2025-07-03

**Rating:** 5
**Confidence:** 4

**Summary:**

The paper introduces a novel benchmark that extends existing 3D scene datasets by combining scene graphs with large language models (LLMs) and human annotators to generate referring expressions at area, space, object, and part levels. The authors evaluate current LLMs, multimodal LLMs (MLLMs), and 3D visual grounding models against human performance. They find that vision-language models typically outperform both pure LLMs and 3D-only approaches—yet even the best model (GPT-4-mini) still lags behind humans on spatial reasoning tasks. To boost visual perception, they propose a pipeline that uses either an LLM or human to select candidate objects from a prompt, then samples video frames where those objects appear most frequently. This yields significant improvements over baselines, especially when humans choose the keyframes. Finally, they enhance relational reasoning by augmenting scene-level bird's-eye-view images with x-y axes and enriching object-level scene graphs with orientation information, further narrowing the gap with human performance.

**Dataset Code Accessibility:**

Yes

**Ethical Considerations:**

No, there are no or only very minor ethics concerns

**Final Justification:**

The authors addressed the concerns I had and their work is in good quality.

**Limitations Weaknesses:**

The authors discuss the possible limitations in Section D of the supplementary material.
I would like also to add:
- Did the authors consider evaluating a 3D grounding LLM as well, such as [1] and [2,] to have a better comparison with the pure LLM and MLLM models?

[1] Yu, Hanxun, et al. "Inst3d-lmm: Instance-aware 3d scene understanding with multi-modal instruction tuning." *Proceedings of the Computer Vision and Pattern Recognition Conference*. 2025.
[2] Chen, Yilun, et al. "Grounded 3d-llm with referent tokens." *arXiv preprint arXiv:2405.10370* (2024).

**Strengths Contributions:**

1. The paper is well-written and logically structured, with clear explanations throughout.
2. The paper provides a thorough cross‐model evaluation, assessing pure LLMs, multimodal LLMs, and 3D visual grounding models against human performance.
3. The authors propose the visual perception and relational reasoning enhancement pipelines to improve the performance of GPT 4.1 on each task level.
4. The authors provide an analysis of the reasoning mistakes in GPT-4.1 answers to highlight the challenge of their data and the shortcomings of current MLLMs.

---

> ### Author Rebuttal · Authors · 2025-07-30
>
> Thank you for the valuable feedback. Below, we provided detailed response to your comments.
>
> > **Q1:** Did the authors consider evaluating a 3D grounding LLM as well, such as [1] and [2] to have a better comparison with the pure LLM and MLLM models?
> >
> > [1] Yu, Hanxun, et al. "Inst3d-lmm: Instance-aware 3d scene understanding with multi-modal instruction tuning." Proceedings of the Computer Vision and Pattern Recognition Conference. 2025. [2] Chen, Yilun, et al. "Grounded 3d-llm with referent tokens." arXiv preprint arXiv:2405.10370 (2024).
>
>
> We greatly appreciate the suggestion to evaluate the two 3D grounding LLMs you mentioned and are actively working on these experiments.
>
> Inst3D-LMM has not released model checkpoints; we have contacted the authors and will conduct evaluation once the checkpoint is provided. For Grounded 3D-LLM, we are currently running the experiments and will provide the evaluation results during the upcoming discussion period.
>
> Also, we will include these results in the revised version of the manuscript.

---

> > ### Author Response · Authors · 2025-08-02
> >
> > We would like to provide evaluation results for Grounded 3D-LLM in response to your concerns. As shown in the following table, we report  Acc@0.25IoU and include the results of the other three 3D visual grounding models reported in our paper(i.e., PQ3D, 3D-VisTA, Chat-Scene in Table 2 of our paper) for comparison. Grounded 3D-LLM demonstrates competitive overall performance among 3D visual grounding models. However, the performance of these models remains limited, particularly at the space level. This observation is consistent with our analysis in lines 188–192 of the paper. We will include the results in the revised manuscript.
> >
> > |                 |  Area Level   | Space Level   | Object Level | Part Level | Overall |
> > | --------------- | --- | --- | -------- | -------- | -------- |
> > | PQ3D            |   30.69   |   7.98  |      24.42     |      20.90     |    17.46      |
> > | 3D-VisTA        |  29.10   |    7.16  |    25.05      |       20.64    |      17.20     |
> > | Grounded 3D-LLM |  49.25   |  6.15   |    26.36      |    19.51      |     20.33     |
> > | Chat-Scene      |   49.10  |    6.12  |     31.73     | 25.31     | 23.41     |

---

> > > ### Comment · Area_Chair_YwzH · 2025-08-05
> > > **Discussion and Rating**
> > >
> > > Hi Reviewer,
> > >
> > > The authors have provided the rebuttal. What are your thoughts on the response? Please engage in the discussion with the authors as soon as possible, as the deadline for discussion is August 8th.
> > >
> > > Thanks,
> > >
> > > AC

---

> > > ### Comment · Reviewer_ZXVe · 2025-08-08
> > >
> > > Thanks to the authors for the rebuttal, which addresses my concerns. I will be keeping my rating of 5.

---

> > > > ### Author Response · Authors · 2025-08-09
> > > >
> > > > Dear Reviewer ZXVe,
> > > >
> > > > &emsp;&emsp; We would like to sincerely thank you again for your valuable feedback, as well as for your recognition and encouragement of our work. We will further improve the manuscript by incorporating these results. If you have any further questions or suggestions, please feel free to reach out to us at any time.

---

### Note · Authors · 2025-08-13

Dear all Reviewers and Area Chair,

We would like to express our gratitude for the time and engagement during the rebuttal and discussion period.

**Key Strengths Recognized by Reviewers:**
1. Anywhere3D-Bench, the first 3D visual grounding benchmark covering four hierarchical levels of 3D visual grounding—area, space, object, and part (`kMLK`, `r58L`, `dPCa`)
2. Comprehensive evaluation across LLMs, MLLMs, and 3D visual grounding specialist models, offering insights into current models' capabilities and limitations (`ZXVe`, `r58L`, `dPCa`)
3. The space-level tasks, requiring reasoning about unoccupied space beyond objects, represent a particularly novel and challenging aspect of the benchmark (`r58L`, `dPCa`)
4. Well-written and logically structured manuscript with rich implementation details to ensure reliability and reproducibility (`ZXVe`, `kMLK`)


**Key Concerns Addressed During Rebuttal**
1. Benchmarked a 3D grounding LLM (*Grounded 3D-LLM*) on Anywhere3D-Bench
2. Provided additional comparison between Anywhere3D-Bench and prior benchmarks, highlighting (1) a unique space-level visual grounding task, (2) a holistic evaluation for multi-level visual grounding in 3D scenes.
3. Demonstrated that Anywhere3D’s size is comparable to existing benchmarks (e.g., ScanReason, SceneFun3D) and outlined concrete strategies, with a dataset expansion plan to further increase its scale.
4. Clarified the potential fairness concern of Qwen-generated captions by comparing models' performance using GPT-4.1 generated captions v.s. Qwen generated captions.
5. Provided quantitative details on the divergence of final expressions from the original expression generated by GPT.

**Planned Revision for Manuscript:**
1. Integrate *Grounded 3D-LLM* evaluation results into main results.
2. Include additional Anywhere3D v.s. existing benchmark comparison table and GPT-4.1 v.s. Qwen caption comparison table.
3. Expand Anywhere3D dataset and report results on the expanded version.
4. Add quantitative details on divergence between GPT-generated expressions and final expressions.

We are encouraged that all reviewers expressed a positive attitude toward our paper(`ZXVe`: remaining rating of 5, `kMLK`: 3 → 4, `r58L`: 4 → 5, `dPCa`: 4 → 5). We will further improve the manuscript by incorporating their valuable suggestions. Thank you again for your valuable feedback and positive recognition.

Best,

Authors

---

### Decision · Program_Chairs · 2025-09-18

**Decision:**

Accept (poster)

**Comment:**

This paper was reviewed by four experts in the field. The recommendations are (Accept x 3, Borderline Accept). Based on the reviewers' feedback, the decision is to recommend the acceptance of the paper. While the reviewers appreciate the contribution of the work, they also raise some valuable concerns. The reviewers collectively highlight several weaknesses in the current work, primarily focusing on the limitations of the dataset size and the perceived lack of novelty in the proposed benchmark. The concerns regarding fairness in evaluation methods and the potential biases introduced by using GPT-4 for expression generation are also notable. To enhance the study, it is recommended that the authors consider integrating a broader range of data, exploring novel methodologies that clearly differentiate their contributions, and addressing the evaluation fairness to ensure a more robust and equitable assessment of model performance. These proposed valuable concerns should be addressed in the final camera-ready version of the paper. The authors are encouraged to make the necessary changes to the best of their ability.